# Leveraging World Model Disentanglement in Value-Based Multi-Agent Reinforcement Learning

## Abstract

In this paper, we propose a novel model-based multi-agent reinforcement learning approach named Value Decomposition Framework with Disentangled World Model to address the challenge of achieving a common goal of multiple agents interacting in the same environment with reduced sample complexity. Due to scalability and non-stationarity problems posed by multi-agent systems, model-free methods rely on a considerable number of samples for training. In contrast, we use a modularized world model, composed of action-conditioned, action-free, and static branches, to unravel the complicated environment dynamics. Our model produces imagined outcomes based on past experience, without sampling directly from the real environment. We employ variational auto-encoders and variational graph auto-encoders to learn the latent representations for the world model, which is merged with a value-based framework to predict the joint action-value function and optimize the overall training objective. Experimental results on StarCraft II micro-management, Multi-Agent MuJoCo, and Level-Based Foraging challenges demonstrate that our method achieves high sample efficiency and exhibits superior performance compared to other baselines across a wide range of multi-agent learning tasks.

## 1 Introduction

As teams of agents expand in size, their potential increases concomitantly. However, the complexity of coordinating the units grows rapidly as more interactions and constraints must be considered. This work proposes to extend the current state-of-the-art approaches, based on latent imagination with world models (Hafner et al., 2022), to multi-agent reinforcement learning (MARL) with the key concept of disentanglement, which facilitates effective scaling to unprecedented multi-agent team sizes and problem complexities.

Recently, model-based reinforcement learning (MBRL) has demonstrated sample efficiency and scalability in handling large single-agent tasks (Xu et al., 2018; Hafner et al., 2020; 2022; Yang & Wang, 2020; Luo et al., 2022). We explore its application to MARL with an emphasis on latent space generative models, as each agent may only have access to local observations, suffering from the partial observability issue. The majority of applied MARL research has been centered around model-free methods (Tampuu et al., 2017; Lowe et al., 2017; Iqbal & Sha, 2018; Zhou et al., 2020; Jeon et al., 2022), while model-based algorithmic studies are still slowly progressing from simple stochastic games to more complex scenarios (Brafman & Tennenholtz, 1999; 2003; Zhang et al., 2021a).

Training model-based policies can be challenging across various domains (Shen et al., 2020). In multi-agent systems, changes occurring around an agent are frequently uncontrollable due to simultaneous interactions with other agents. This leads to a complex learning problem for holistic models, while a recent study on modular representation attempted to decouple the environment dynamics into passive and active components (Pan et al., 2022). Inspired by their work, we maintain three branches within our world model: action-conditioned, action-free, and static. Each branch tackles a unique learning problem, including grasping common interaction-free (static) state features, identifying passive (action-free) forces surrounding an agent, and understanding the full complexity of the active (action-conditioned) control system. The predictions of the branches are synergistic and produce informative latent space roll-outs, which are learned by variational

and graph convolutional auto-encoders. Combined with individual action values from the joint agent network, the simulated roll-outs are passed into a mixing network, which adopts value decomposition techniques for estimating joint value functions. To the extent of our knowledge, our study marks the first endeavor to leverage the disentanglement of latent representations in MARL.

We evaluate our method on well-known MARL environments including StarCraft II MARL benchmark (SMAC) (Samvelyan et al., 2019), Multi-Agent MuJoCo (MAMuJoCo) (Peng et al., 2021), and Level-Based Foraging (LBF) (Christianos et al., 2020). We have constantly observed the performance of our approach matching or surpassing the state-of-the-art across a rich variety of tasks. The improvement in episodic returns is evident in MAMuJoCo and LBF test beds. Moreover, our method consistently develops winning strategies for *Super-Hard* SMAC challenges, a group of scenarios characterized by a high diversity of unit types involved in complicated interactions. This reinforces the importance of our disentanglement approach for large-scale MARL tasks.

## 2 Related Work

### 2.1 Model-Based Reinforcement Learning

A fundamental concept in MBRL is the environment model (Luo et al., 2022). It is an abstraction of the environment dynamics, formulated as a Markov decision process (MDP). The agent has the ability to imagine, which means it can generate simulated samples in the environment model. This reduces the interactions with the real environment and improves sample efficiency (Xu et al., 2018; Janner et al., 2019).

As a class of the environment model, world models can be applied to learning representations and behaviors (Watter et al., 2015; Pan et al., 2022). Built on the framework of PlaNet (Hafner et al., 2019), Dreamer (Hafner et al., 2023) uses a world model to train the agent with latent imaginations, efficiently predicting long-term behaviors. These methods optimize the policy by directly learning from the model-based imaginary experiences, in a style analogous to the well-known Dyna algorithm (Sutton, 1991).

On the other hand, the agent can also extract useful knowledge from simulations performed by the environment model, instead of relying solely on the imagined data. For example, incentivizing scalable and efficient exploration of agents in complex domains has always been a challenge in RL. An effective approach involves designing the exploration strategy based on bonuses derived from a learned system dynamics model or the agent's belief of the dynamics model (Stadie et al., 2015; Houthooft et al., 2016). Gu et al. (2016) discover that iteratively refitting local linear models to local on-policy roll-outs can substantially accelerate off-policy algorithms, such as Q-learning. Racaniere et al. (2017) introduce the Imagination-Augmented Agent (I2A), which consists of a model-based path and a model-free path. The policy network combines the information from both paths and outputs the imagination-augmented policy. Liu et al. (2019) improves I2A by generating more diverse and informative trajectories to help with the agent's performance while maintaining the consistency of the trajectories with the real world. PMA-DRL (Luo & Wang, 2020) is another model-augmented framework under the parallel RL setting, in which multiple agents are involved but each of them interacts with its own environment instance (Nair et al., 2015; Mnih et al., 2016). Each agent in PMA-DRL is accompanied by a local dynamics model (LDM) to facilitate exploration and an ensemble of LDMs is applied to reduce the model bias. Li & Lasenby (2024) approach the Variable Speed Limit optimization problem by constructing an imagination-augmented framework, showing that it offers a promising potential to raise motorway productivity and ameliorate traffic conditions.

### 2.2 Multi-Agent Model-Based Reinforcement Learning

The development of multi-agent MBRL has mainly focused on theoretical analyses (Bai & Jin, 2020; Zhang et al., 2020; Yang & Wang, 2020). Existing algorithms in this field rely heavily on specific prior knowledge such as information on states and adversaries (Brafman & Tennenholtz, 2003; Park et al., 2019b; Zhang et al., 2021b), which can be inaccessible in many scenarios. Wang et al. (2022) review recent advancements in theoretical analyses and methods within multi-agent MBRL. This paper categorizes the methods based on three dimensions: training schemes, opponent awareness, and environment usage. It outlines the strengths

and weaknesses of these algorithms in centralized and decentralized training schemes. Meanwhile, it remarks that the algorithms in this domain still remain underexplored. Pasztor et al. (2021) propose M³-UCRL, integrating model-based RL with mean-field game theory and deriving theoretical regret bounds via a mean-field-based analysis. M³-UCRL is applicable to mean-field control problems with unknown system dynamics. However, the limitations of mean-field theory restrict the suitability of the method to large systems with many agents only. MAMBA (Egorov & Shpilman, 2022) treats the environment model as an instance of communication, using direct messages to assist with decentralized execution and account for the limitation on message channel bandwidth. The computation time per collected trajectory for MAMBA is nonetheless slower than previous MARL methods since it demands trajectories with very long roll-out horizons to learn the policy. Sessa et al. (2022) introduce H-MARL, constructing statistical confidence bounds around the unknown environment transition and using them to create a hallucinated optimistic game for agents. Although H-MARL is theoretically proven to balance exploration and exploitation, it is only tested in a simulated driving scenario and lacks experiments in diverse environments with intractable transition functions.

Yang & Wang (2020) and Zhang et al. (2020) review the complexities of MARL algorithms arising from non-stationarity, partial observability, and agent coordination issues. The authors consider model-based MARL as a promising approach to improve sample efficiency and model effectiveness, and believe that it deserves more attention from MARL researchers. Luo et al. (2022) provide a comprehensive survey of model-based RL, highlighting the importance of model learning, planning, and integration with model-free methods. The survey suggests that potential directions in the development of model-based MARL include the new design of decentralized methods and communication protocols based on the learned models. Zhang et al. (2021b) introduce a model-based multi-agent policy optimization framework. By learning a model of the adversaries' behaviors and adjusting the rollout strategy based on the learned model, the approach improves the efficiency of policy optimization in competitive settings. Zhang et al. (2020) propose a model-based MARL algorithm for zero-sum Markov games, achieving near-optimal sample complexity. The algorithm leverages the game structure to learn a model of the opponent's strategy and update the agent's policy accordingly. The authors provide theoretical guarantees on the sample complexity, proving that it is lower than the complexity of model-free methods. Park et al. (2019b) investigate MBRL in competitive multi-opponent games. The authors design an approximate model learning framework that estimates the transition dynamics and rewards of the game environment using auxiliary networks. The communication between agents is promoted by policy gradients in actor-critic networks.

However, both (Zhang et al., 2021b) and (Zhang et al., 2020) mainly concentrate on the theoretical studies of MARL algorithms and have limitations in application. The proposed approach in (Zhang et al., 2021b) requires prior knowledge about opponents to construct the opponent model, which may cause generalization errors that are hard to estimate. Zhang et al. (2020) only study the setting of zero-sum Markov games. Park et al. (2019b) use an auxiliary prediction network, which may lead to prediction errors in complex environments. Moreover, Park et al. (2019b) conducted experiments in only one benchmark and compared against only one baseline (MADDPG). In contrast, our VDFD method is tested across different benchmark environments and compared with many MARL baselines, demonstrating high efficiency and generalization capability even in complex settings. VDFD estimates the prior and posterior distributions of the states under partial observability, without the need for explicit opponent modeling.

## 2.3 Value-Based MARL Methods

A large class of RL algorithms applied in multi-agent systems are value-based methods. They typically compute value function estimates and differ in the extent of centralization. In a fully decentralized scenario (Tan, 1997; Tampuu et al., 2017), each agent improves its own policy with the assumption of participating in a stationary environment. Because each agent fails to account for the behaviors selected by other agents or the rewards received by them, the assumption of environment stationarity in traditional RL no longer holds (Yang & Wang, 2020). There is no theoretical guarantee that an independent learning algorithm will converge in this case (Zhang et al., 2021a).

On the other hand, a centralized controller gathers the observation and the joint action of the agents. A method in which all agents have access to global state information and are aware of the non-stationarity in the environment also demonstrates centralization. Boutilier (1996) described the multi-agent Markov decision

process, assuming that all agents observe the global rewards. Guestrin et al. (2002) designed a global payoff function as the sum of local payoff functions, which are a variant of local rewards. The global payoff function was optimized by agents using a variable elimination algorithm. However, due to the combinatorial nature of MARL, the scale of the joint action space will expand exponentially with respect to the number of agents within the same environment (Kok & Vlassis, 2006; Zhang et al., 2021a). As a result, appropriate remedies for scalability issues need to be found.

Recent studies focused on MARL algorithms that lie between the two extremes of decentralization (Yang et al., 2019; Son et al., 2019; Rashid et al., 2020; Wang et al., 2020a;b;c; Jeon et al., 2022), according to the paradigm of Centralized Training with Decentralized Execution (CTDE) (Kraemer & Banerjee, 2016). CTDE stipulates that agents are allowed to exchange information with other agents only during training and they must act in a decentralized manner during execution. Following this paradigm, the value decomposition network (VDN) (Sunehag et al., 2017) shares network weights and information channels and specifies roles across them. VDN leverages the joint value function $Q_{tot}$ of the learning agents, which can be additively factorized into individual Q functions $\tilde{Q}_i$. The assumption of VDN can be overly restrictive, since the additive value decomposability may not hold for more complex action-value functions. QMIX (Rashid et al., 2018) replaces the full factorization in VDN with the enforcement of monotonicity between the joint $Q_{tot}$ and the individual $\tilde{Q}_i$, which enables it to represent a larger class of action-value functions than VDN. Jianye et al. (2023) devise a unified multi-agent permutation framework that leverages the permutation invariance (PI) and permutation equivariance (PE) inductive biases for state space reduction. While the empirical evaluations show that this framework can be combined with existing algorithms like QMIX to enhance learning efficiency, it remains an open question whether this framework can be applied in scenarios where the structural information is unavailable or the observations are images.

In our approach, a value factorization framework is employed to mix the agents' individual action values. The framework receives the outputs from the disentangled representation learning process, in which the future states are inferred using informative simulated roll-outs. It can therefore predict real-world dynamics and approximate the global value function with high accuracy.

## 3 Background

### 3.1 Dec-POMDP

When agents engage in a task, it is possible that each of them has a limited field of view. The decentralized partially observable Markov decision process (Dec-POMDP) is appropriate for modeling collaborative agents in a partially observable environment (Oliehoek & Amato, 2016).

**Definition 3.1.** A Dec-POMDP is defined by a tuple

$$M := \langle \mathcal{S}, \mathcal{A}, \mathcal{N}, T, \mathcal{Z}, O, R, \gamma \rangle,$$

where $\mathcal{S}$ is the state space of all agents, $\mathcal{A}$ is the joint action space of all agents, $\mathcal{N} = \{1, ..., N\}$ represents the set of $N$ agents, $T$ is the state transition function, $\mathcal{Z}$ represents the observation space, $O$ is the observation function, $R$ is the reward function, and $\gamma \in [0, 1]$ represents the discount factor with respect to time.

At time step $t$, each agent $i \in \mathcal{N}$ chooses an action $a^i$ from its own action space $\mathcal{A}^i$ to form the joint action $\mathbf{a} \in \mathcal{A}$, where $\mathbf{a} := (a^1, ..., a^N)$ and $\mathcal{A} := \times_{i \in \mathcal{N}} \mathcal{A}^i$. Then the environment moves from $s$ to $s'$ based on $T(s'|s, \mathbf{a}) : \mathcal{S} \times \mathcal{A} \times \mathcal{S} \rightarrow [0, 1]$. Every agent $i$ draws an observation $z \in \mathcal{Z}$ according to $O(s, i) : \mathcal{S} \times \mathcal{N} \rightarrow \mathcal{Z}$ because of partial observability. $i$ has its own action-observation history, denoted by $\tau^i \in \mathcal{T}^i := (\mathcal{Z} \times \mathcal{A}^i)^*$, and selects $a^i$ by its policy $\pi^i(a^i|\tau^i) : \mathcal{T}^i \times \mathcal{A}^i \rightarrow [0, 1]$. The learning goal is to maximize the expected return by optimizing the joint policy $\pi = (\pi^1, ..., \pi^N)$. The joint action-value function of $\pi$ is

$$Q^\pi(s_t, \mathbf{a}_t) = \mathbb{E}_{s_{t+1:\infty}, \mathbf{a}_{t+1:\infty}}[G_t|s_t, \mathbf{a}_t],$$

where $G_t = \sum_{k=0}^\infty \gamma^k r_{t+k}$ is the discounted return; $r_{t+k}$ is the reward computed by $R$ for all agents at time step $t + k$. The reward function $R : \mathcal{S} \times \mathcal{A} \rightarrow \mathbb{R}$ maps the states and joint actions to real numbers. $R$ can be

considered as a function that outputs the immediate reward for each joint action $\mathbf{a} \in \mathcal{A}$. In particular, the reward $r_{t+k}$ can also be written as $R(s_{t+k}, \mathbf{a}_{t+k})$. Therefore, $r_{t+k}$ depends on the state $s_{t+k}$ and the joint action of all agents $\mathbf{a}_{t+k}$.

### 3.2 Value Decomposition and Individual-Global-Max

Based on the CTDE paradigm, value decomposition is an effective technique deployed in MARL as it encourages collaboration between agents (Son et al., 2019). To apply this technique, we need to define the condition of Individual-Global-Max (IGM):

**Definition 3.2.** Let $\mathcal{A}$ be the joint action space and $\mathcal{T}$ be the joint action-observation history space. Denote the agents' joint action-observation histories as $\boldsymbol{\tau} \in \mathcal{T}$ and their joint actions as $\mathbf{a} \in \mathcal{A}$. Given the joint action-value function $Q_{tot} : \mathcal{T} \times \mathcal{A} \to \mathbb{R}$, if there exist individual $\{Q_i : \mathcal{T}^i \times \mathcal{A}^i \to \mathbb{R}\}_{i \in \{1,...,N\}}$ such that the following holds:

$$\arg\max_{\mathbf{a} \in \mathcal{A}} Q_{tot}(\boldsymbol{\tau}, \mathbf{a}) = \begin{bmatrix} \arg\max_{a^1} Q_1(\tau^1, a^1) \\ \vdots \\ \arg\max_{a^N} Q_N(\tau^N, a^N) \end{bmatrix} \tag{1}$$

then $\{Q_i\}_{i \in \{1,...,N\}}$ satisfy the Individual-Global-Max (IGM) condition for $Q_{tot}$ with $\boldsymbol{\tau}$, which means $Q_{tot}(\boldsymbol{\tau}, \mathbf{a})$ can be decomposed by $\{Q_i\}_{i \in \{1,...,N\}}$.

In order to guarantee that the IGM condition holds, different assumptions have been made. VDN (Sunehag et al., 2017) utilizes a sufficient condition, named *additivity*, for IGM:

$$Q_{tot}(\boldsymbol{\tau}, \mathbf{a}) = \sum_{i=1}^{N} Q_i(\tau^i, a^i)$$

VDN is able to factorize the joint value function assuming the additivity of individual value functions. Alternatively, QMIX (Rashid et al., 2018) uses a sufficient condition called *monotonicity* for IGM and proves that IGM is guaranteed under this assumption:

$$\frac{\partial Q_{tot}(\boldsymbol{\tau}, \mathbf{a})}{\partial Q_i(\tau^i, a^i)} \geq 0, \forall i \in \{1, ..., N\}.$$

Lastly, QTRAN (Son et al., 2019) proposes to find individual action-value functions $[Q_i]$ that factorize the original joint action-value function $Q_{jt}$ and to transform $Q_{jt}$ into a new value function $Q'_{jt}$ that shares the same optimal joint action with $Q_{jt}$. The sufficient condition for $[Q_i]$ to satisfy the IGM is described in detail in Theorem 1 of (Son et al., 2019).

IGM indicates that a MARL task can be solved in a decentralized manner as long as local and global action-value functions are consistent (Sunehag et al., 2017; Rashid et al., 2018; Son et al., 2019).

### 3.3 Structured Variational Inference

The process of structured variational inference involves approximating some complicated distribution $p(\mathbf{y})$ with $q(\mathbf{y})$, another potentially simpler distribution (Levine, 2018). The approximate inference is performed by optimizing the variational lower bound. Huang et al. (2020) introduce a probabilistic graphical model (PGM) for single-agent partially observable settings and solve the variational inference problem under the model. The variational lower bound associated with the PGM can be written as follows:

$$\log p(\mathcal{O}_{0:T}, a_{0:T}, z_{1:T}) = \log \mathbb{E}_{q_\theta(s_{1:T}|\mathcal{O}_{1:T}, a_{0:T}, z_{1:T})} \left[ \frac{p(s_{1:T}, \mathcal{O}_{0:T}, a_{0:T}, z_{1:T})}{q_\theta(s_{1:T}|\mathcal{O}_{0:T}, a_{0:T}, z_{1:T})} \right]$$

$$\geq \mathbb{E}_{q_\theta(s_{1:T}|\mathcal{O}_{1:T}, a_{0:T}, z_{1:T})} \log \left[ \frac{p(s_{1:T}, \mathcal{O}_{0:T}, a_{0:T}, z_{1:T})}{q_\theta(s_{1:T}|\mathcal{O}_{0:T}, a_{0:T}, z_{1:T})} \right],$$

where $s, a, z$ are the state, action, and observation of the agent, $q_\theta$ is the approximate function, and $\theta$ stands for the learnable parameter. $\mathcal{O}_t$ is a binary random variable introduced by Levine (2018) related to maximum entropy reinforcement learning. It indicates the optimality of the action at time $t$.

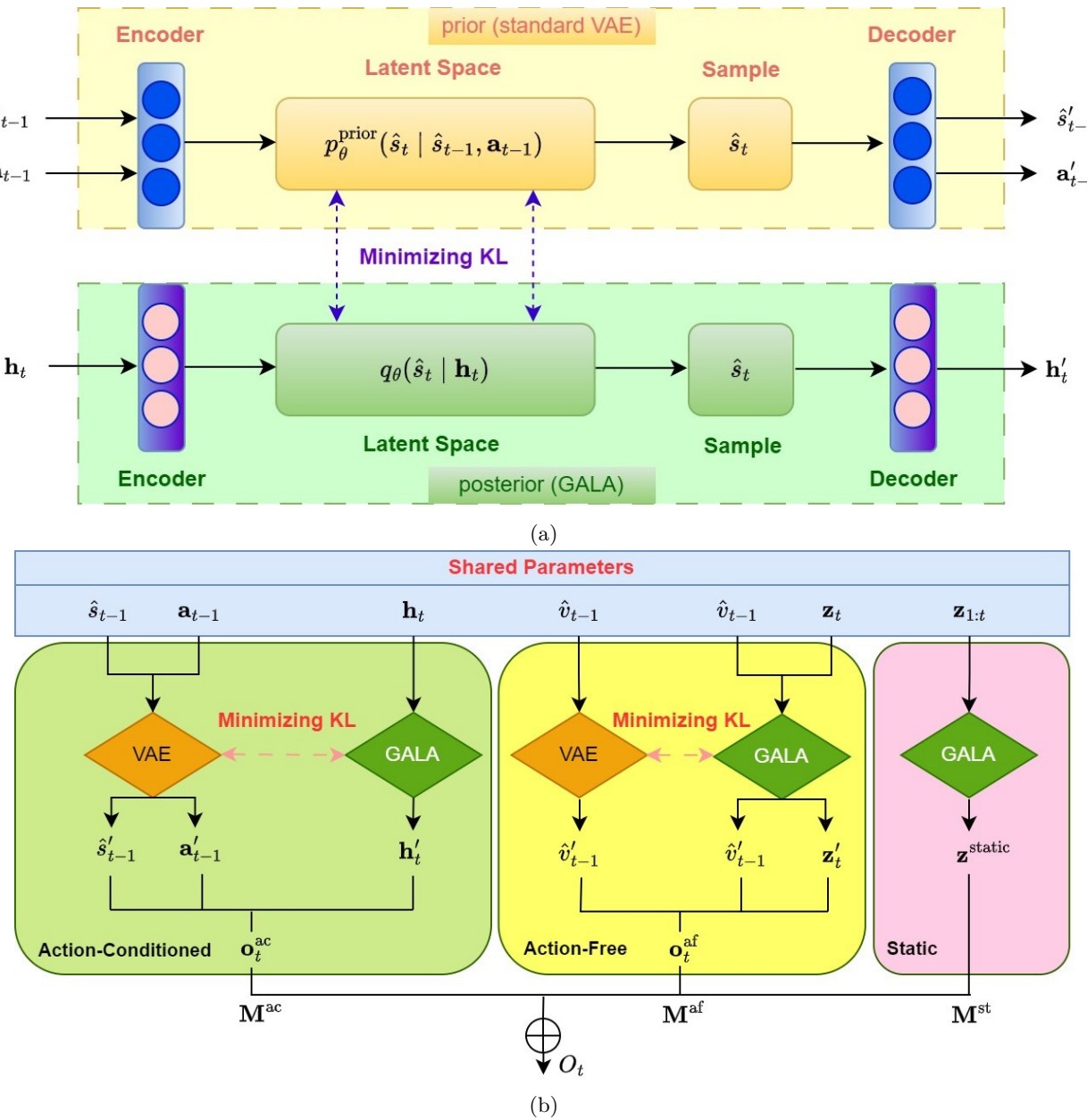

(a)

(b)

Figure 1: The learning processes and the structure of the modularized world model. (a) Using VAE and GALA to learn the prior and the posterior in the KL term. The KL distance between the distributions connected by the purple dotted line is minimized. $\mathbf{h}_t$ can be viewed as a function of $\hat{s}_{t-1}, \mathbf{a}_{t-1}$, and $\mathbf{z}_t$. (b) The three modules of the world model with shared parameters. In the action-conditioned branch, the input to GALA is $\mathbf{h}_t$. In the action-free branch, the input to GALA is $\{\hat{v}_{t-1}, \mathbf{z}_t\}$. In the static branch it is $\mathbf{z}_{1:t}$.

## 4   Method

When multiple agents are interacting with each other simultaneously, the environment dynamics can be more sophisticated than in single-agent scenarios. To learn latent dynamics models effectively, we can leverage disentangled representation learning (Goyal et al., 2021; Pan et al., 2022). We hereby introduce a novel model-based multi-agent reinforcement learning algorithm named **V**alue **D**e-composition **F**ramework with **D**isentangled World Model (VDFD). Specifically, the world model is decomposed into three modules: an

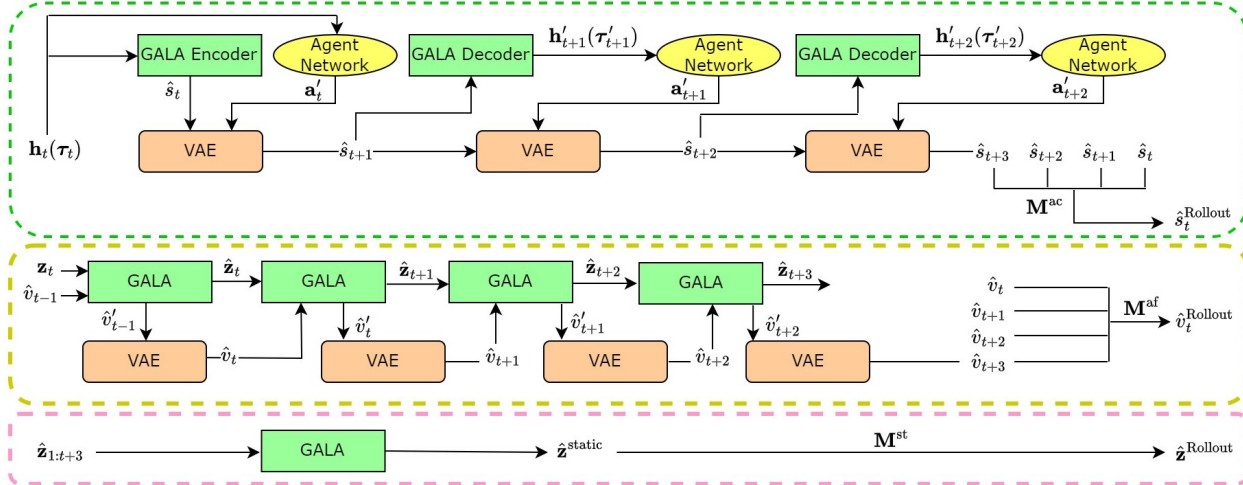

Figure 2: An illustration of the modularized world model when the roll-out horizon $j$ is set to 3. The modules enclosed by green, yellow, and pink dashed lines represent the action-conditioned, action-free, and static branches, respectively. The bright yellow ellipse stands for the agent network, which implements the joint policy for all agents. The joint actions can therefore be extracted from the agent network. We link the controllable latent states $\{\hat{s}_t, \ldots, \hat{s}_{t+3}\}$ together to show that they are aggregated and transformed into $\hat{s}_t^{\text{Rollout}}$ via the mask $\mathbf{M}^{\text{ac}}$. This is true for the non-controllable latent states $\{\hat{v}_t, \ldots, \hat{v}_{t+3}\}$ as well.

action-conditioned (controllable) branch that responds to and depends on the actions of the agents, an action-free (non-controllable) branch that is independent of the agents' behaviors, and a static branch that consists of environmental features and remains unchanged.

In this section, we analyze how the variational lower bound for the latent state inference in the world model can be deduced and optimized when the environment is modeled as Dec-POMDP. We then discuss the details of world model disentanglement. Next, we elaborate on the entire workflow of VDFD. It consists of two parts, world model imagination and reinforcement learning with value function factorization. For the first part, the controllable and non-controllable branches of the world model infer simulated roll-out states, while the static branch reconstructs the observations. We employ a mixing network proposed by Rashid et al. (2018) for the second part. It receives the actual global state and the individual functions $\{Q_1, ..., Q_N\}$ provided by the agent network. Moreover, the simulated roll-outs from the decoupled world model are concatenated and passed into the mixing network, contributing to the computation of the global action-value function. Lastly, we derive the overall learning objective by summing up the loss functions of different components.

## 4.1 Disentangling the World Model for Representation Learning

Since the global state $s$ is inaccessible to agents outside the centralized training phase, we need to infer the latent state from local actions $\mathbf{a}$ and observations $\mathbf{z}$. We implement a variational auto-encoder (VAE) and a variational graph auto-encoder (VGAE) to approximate the prior and posterior distributions of $\hat{s}$, which is a crucial step in maximizing the evidence lower bound for Dec-POMDP.

To learn compact and accurate representations of a multi-agent environment, we disentangle dynamic and static components, denoted as $d$ and $\mathbf{z}^{\text{static}}$, from the world model and train them jointly. The mixed dynamics $d$ of the model[1] can be decoupled into an action-conditioned branch, which corresponds to the state transition $\hat{s}_t \sim q(\cdot|\hat{s}_{t-1}, \mathbf{a}_{t-1}, \mathbf{z}_t)$, and an action-free branch, which is beyond the control of the agents and can therefore be separated from their joint actions. The non-controllable transition can be modeled as $\hat{v}_t \sim q(\cdot|\hat{v}_{t-1}, \mathbf{z}_t)$. It is worth noting that the controllable latent state $\hat{s}$ is conceptually distinct from the real global state $s$, and the static component $\mathbf{z}^{\text{static}}$ is conceptually distinct from the joint observation $\mathbf{z}$, despite their similar notations. The roll-outs from the decoupled branches are jointly conveyed to the value

---

[1]We will revisit the concept of $d$ in Section 4.1.4.

factorization framework for centralized training. In the following subsections, we first present the analysis of the variational lower bound with respect to the controllable dynamics, which involve the state transitions of $\hat{s}$. Then we demonstrate how the decoupled world model can be integrated into the framework of VDFD. Finally, we examine the learning objectives that embody the model disentanglement in detail.

### 4.1.1  Deriving the Variational Lower Bound

To perform latent space inference in Dec-POMDP, we need to deduce the evidence lower bound (ELBO) corresponding to this setting. Inspired by the probabilistic approach in (Huang et al., 2020), we create two approximate functions $q_\pi(\cdot)$ and $q_\theta(\cdot)$, where $\theta$ represents the learnable parameter, $q_\pi(\cdot)$ is used for approximating the optimal joint policy, and $q_\theta(\cdot)$ is the inference function for latent states[2]. When we keep $q_\theta(\cdot)$ fixed, $q_\pi(\cdot)$ can be trained using soft Q-learning or vanilla Q-learning. When we fix $q_\pi(\cdot)$ as the optimal policy, $q_\theta(\cdot)$ can be learned for the latent space. Given time steps $t \in [0, T]$, joint actions $\mathbf{a} \in \mathcal{A}$ and joint observations $\mathbf{z} \in \mathcal{Z}$, we can define the approximate posterior as $q_\theta(\hat{s}_t | \hat{s}_{t-1}, \mathbf{a}_{t-1}, \mathbf{z}_t)$. This function is used to infer the latent states. $\hat{s}_t$ is an abstract representation of the agents' local observations, which indicates that $\mathbf{z}_t \sim p(\mathbf{z}_t | \hat{s}_t)$. We can then derive the ELBO of Dec-POMDP, $\mathcal{L}_{\text{ELBO}}(\mathbf{a}_{0:T}, \mathbf{z}_{1:T})$, as below. The full deduction can be found in Appendix E.

$$
\begin{aligned}
\mathcal{L}_{\text{ELBO}}(\mathbf{a}_{0:T}, \mathbf{z}_{1:T}) &= \log p(\mathbf{a}_{0:T}, \mathbf{z}_{1:T}) \\
&= \log \mathbb{E}_{q_\theta(\hat{s}_{1:T} | \mathbf{a}_{0:T}, \mathbf{z}_{1:T})} \left[ \frac{p(\hat{s}_{1:T}, \mathbf{a}_{0:T}, \mathbf{z}_{1:T})}{q_\theta(\hat{s}_{1:T} | \mathbf{a}_{0:T}, \mathbf{z}_{1:T})} \right] \\
&\geq \mathbb{E}_{q_\theta(\hat{s}_{1:T} | \mathbf{a}_{0:T}, \mathbf{z}_{1:T})} \log \left[ \frac{p(\hat{s}_{1:T}, \mathbf{a}_{0:T}, \mathbf{z}_{1:T})}{q_\theta(\hat{s}_{1:T} | \mathbf{a}_{0:T}, \mathbf{z}_{1:T})} \right] \\
&\approx \sum_{t=1}^{T} \{ \log [p(\mathbf{a}_t | \mathbf{z}_t)] + \log [p(\mathbf{z}_t | \hat{s}_t)] - \mathcal{D}_{\text{KL}} [q_\theta(\hat{s}_t | \hat{s}_{t-1}, \mathbf{a}_{t-1}, \mathbf{z}_t) \parallel p(\hat{s}_t | \hat{s}_{t-1}, \mathbf{a}_{t-1})] \} \quad (2)
\end{aligned}
$$

### 4.1.2  Optimizing the ELBO

The $\mathcal{L}_{\text{ELBO}}$ that we obtain is distinct from the variational lower bound in Section 3.3 in two aspects. Firstly, our model is not relevant to maximum entropy reinforcement learning and $\mathcal{L}_{\text{ELBO}}$ does not include $\mathcal{O}_t$ as an extra variable. Secondly, we derive the ELBO with respect to the joint action $\mathbf{a}$ and joint observation $\mathbf{z}$ to solve the variational inference problem in multi-agent settings modeled as Dec-POMDP. In comparison, Huang et al. (2020) only study the POMDP in single-agent domains.

We investigate each term in the sum of Eq.2 to maximize the ELBO. The first term, $\log [p(\mathbf{a}_t | \mathbf{z}_t)]$, stands for the joint policy that is independent of the state inference. Therefore, it can be considered as unrelated to the optimization of the ELBO. The second term, $\log [p(\mathbf{z}_t | \hat{s}_t)]$, indicates that the latent states contain the information from which the local observations can be derived. The last term in $\mathcal{L}_{\text{ELBO}}(\mathbf{a}_{0:T}, \mathbf{z}_{1:T})$ is

$$
\mathcal{D}_{\text{KL}} [q_\theta(\hat{s}_t | \hat{s}_{t-1}, \mathbf{a}_{t-1}, \mathbf{z}_t) \parallel p(\hat{s}_t | \hat{s}_{t-1}, \mathbf{a}_{t-1})],
$$

which denotes the negative Kullback-Leibler (KL) divergence. This term implies that the KL distance between the approximates of posterior and prior should be minimized in the optimization process. As the actual prior distribution $p(\hat{s}_t | \hat{s}_{t-1}, \mathbf{a}_{t-1})$ is unknown, we introduce a generative model $p_\theta^{\text{prior}}(\hat{s}_t | \hat{s}_{t-1}, \mathbf{a}_{t-1})$ to estimate the prior.

### 4.1.3  Using Generative Models

We apply generative models to learning $p_\theta^{\text{prior}}(\hat{s}_t | \hat{s}_{t-1}, \mathbf{a}_{t-1})$ and $q_\theta(\hat{s}_t | \hat{s}_{t-1}, \mathbf{a}_{t-1}, \mathbf{z}_t)$ in the KL term. We construct a VAE for the prior $p_\theta^{\text{prior}}(\cdot)$. It takes in the past state, $\hat{s}_{t-1}$, and the past actions of all agents, $\mathbf{a}_{t-1}$, to compute the prior distribution of the current state. The prior latent state is denoted as $\hat{s}_t \sim p_\theta^{\text{prior}}(\hat{s}_t \mid \hat{s}_{t-1}, \mathbf{a}_{t-1})$.

---

[2]The approximate functions $q_\pi$ and $q_\theta$ should not be confused with the intractable true posterior $q$.

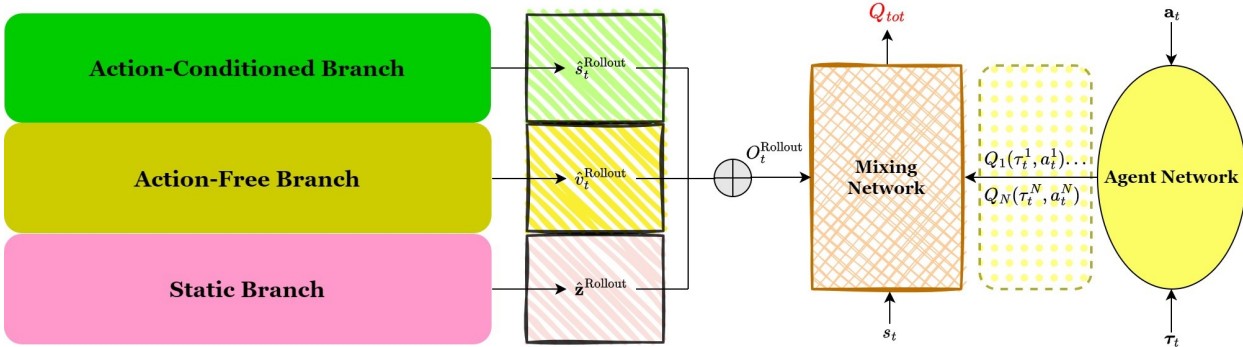

Figure 3: The overall architecture of the VDFD model. The internal structures of the action-conditioned, action-free, and static branches are demonstrated in Figure 2. The inputs to the mixing network include $O_t^{\text{Rollout}}$ from the modularized world model, the global state $s_t$, and $\{Q_1, Q_2, \ldots, Q_N\}$ from the agent network. Because VDFD may need to operate in multi-agent settings with potentially large numbers of agents, we have disabled the calculation of gradients along the simulated roll-outs during training. This allows us to reduce unnecessary memory consumption for computations like gradient back-propagation in the process of world model learning. Moreover, the model and the agent network can be updated asynchronously in practice. This is because the size of the agent network can be very large in certain environments, where the number of agents can exceed 50. The agent network will be updated after a certain number of episodes. The training of the model and the agents can be considered as interleaved.

We employ a variant of the VGAE named GALA (Park et al., 2019a) for the posterior $q_\theta(\cdot)$ to improve the computational efficiency of state inference. GALA is a completely symmetric auto-encoder that combines traditional VAE structure with a Graph Convolutional Network (GCN) encoder (Kipf & Welling, 2016). Additionally, it has a special decoder that performs Laplacian sharpening, which is a counterpart to the encoder that conducts Laplacian smoothing. Unlike previous VGAEs that only use an affinity matrix of the GCN in the decoding phase, GALA is able to directly reconstruct the feature matrix of the nodes. Consequently, it captures the information on the relationship between nodes. Because GALA leverages GCN, it ensures that the number of parameters to be trained remains constant so long as the feature dimension of the nodes remains stable, irrespective of the growth in the number of agents.

We use recurrent neural networks (RNNs) to implement the agent network in Dec-POMDP. We denote the hidden outputs of the RNN for agent $i$ and for the whole network as $h_t^i$ and $\mathbf{h}_t$, respectively. We interpret $h_t^i$ as the integration of all past knowledge specific to agent $i$ and assume that $\mathbf{h}_t$ collectively encapsulates all the past information of the environment. By this assumption, we can reformulate the approximate posterior $q_\theta(\cdot|\hat{s}_{t-1}, \mathbf{a}_{t-1}, \mathbf{z}_t)$ as $q_\theta(\cdot|\mathbf{h}_t)$. Specifically, we use a neural network $f_\theta(\hat{s}_{t-1}, \mathbf{a}_{t-1}, \mathbf{z}_t)$ to extract low dimensional features $\omega_t$, i.e., $\omega_t = f_\theta(\hat{s}_{t-1}, \mathbf{a}_{t-1}, \mathbf{z}_t)$. Then we employ an RNN to encapsulate the past information of the agents, i.e., $\mathbf{h}_t = rnn(\omega_t, \mathbf{h}_{t-1})$ and $\mathbf{h}_0 = \hat{s}_0$. This formulation also holds for each $h_t^i$.

Initially, the posterior latent state is $\hat{s}_t \sim q_\theta(\hat{s}_t|\hat{s}_{t-1}, \mathbf{a}_{t-1}, \mathbf{z}_t)$. With the reparameterization, it can be transformed into $\hat{s}_t \sim q_\theta(\hat{s}_t|\mathbf{h}_t)$. Figure 1a illustrates the learning behaviors of the generative models.

### 4.1.4 Modules of the World Model

We expatiate on our own version of world model disentanglement. At time step $t$, we define the environment dynamics as $d_{1:t}$. By dividing $d_{1:t}$ into a branch of action-conditioned latent states $\hat{s}_{1:t}$ and a branch of action-free latent states $\hat{v}_{1:t}$, we aim at analyzing interactions and understanding relationships between them. We apply the aforementioned generative models to them, as shown in Figures 1a and 1b. Within the action-conditioned branch, we have the prior representations modeled as $\hat{s}_t \sim p_\theta^{\text{prior}}(\hat{s}_t|\hat{s}_{t-1}, \mathbf{a}_{t-1})$ and the posteriors as $\hat{s}_t \sim q_\theta(\hat{s}_t|\mathbf{h}_t)$. We seek to minimize the KL divergence between them. Inside the action-free branch, we model the priors as $\hat{v}_t \sim p_\theta^{\text{prior}}(\hat{v}_t|\hat{v}_{t-1})$ and the posteriors as $\hat{v}_t \sim q_\theta(\hat{v}_t|\hat{v}_{t-1}, \mathbf{z}_t)$, minimizing the KL distance as well. Lastly, the static branch incorporates the observations denoted as $\mathbf{z}_{1:t}$. Because it has no priors or posteriors, we only use GALA for representation learning. Denote the outputs of action-

conditioned, action-free, and static branches as $\mathbf{o}_t^{\mathrm{ac}}$, $\mathbf{o}_t^{\mathrm{af}}$, and $\mathbf{z}^{\mathrm{static}}$, respectively. Define $\odot$ as Hadamard product. Let $\mathbf{M}^{\mathrm{ac}}$, $\mathbf{M}^{\mathrm{af}}$, and $\mathbf{M}^{\mathrm{st}}$ be real-valued masks, which can be viewed as hyperparameters. We obtain the masked output:

$$O_t = \mathbf{M}^{\mathrm{ac}} \odot \mathbf{o}_t^{\mathrm{ac}} + \mathbf{M}^{\mathrm{af}} \odot \mathbf{o}_t^{\mathrm{af}} + \mathbf{M}^{\mathrm{st}} \odot \mathbf{z}^{\mathrm{static}} \tag{3}$$

We can decide whether the action-free module has an impact on learning using our prior knowledge of the tasks. When non-controllable dynamics can be treated as irrelevant time-varying noises, $\hat{v}$ will be zeroed out and the joint policy is merely correlated with $\hat{s}$. For tasks like video prediction and multi-agent cooperation, however, the action-free latent states can affect the decision-making of the agent. The policy and the reward then depend on both $\hat{s}$ and $\hat{v}$.

It is important to note that the non-controllable dynamics $\hat{v}$ may still impact the controllable dynamics $\hat{s}$. For example, when the model is trained in StarCraft II scenarios, the non-controllable branch collects the observations of opponents as data. The attack of the opponents may force the agents to change their strategy to reduce damage. We have considered this possibility, and therefore we propose using the past environmental information, $\mathbf{h}_t$, as input for both the GALA model and the agent network in the action-conditioned branch, as illustrated in Figure 2. By definition, $\mathbf{h}_t$ encapsulates the non-controllable information as well. During non-controllable state transition, GALA outputs $\hat{s}_t \sim q_\theta(\cdot|\mathbf{h}_t)$ and the agent network outputs $\mathbf{a}_t' \sim \pi(\cdot|\boldsymbol{\tau}_t)$. Both $\hat{s}_t$ and $\mathbf{a}_t'$ depend on $\mathbf{h}_t$, which indeed encapsulates the non-controllable factor $\hat{v}$.

In complicated multi-agent systems, the boundary between action-free and action-conditioned components might be unclear. For example, the posterior distribution in the non-controllable branch is $q_\theta(\hat{v}_t|\hat{v}_{t-1}, \mathbf{z}_t)$, but $\mathbf{z}_t$ might be affected by actions of the agents in environments like StarCraft II. Learning to disentangle the dynamics in such systems has always been a challenging research topic in model-based RL. To address this challenge, we have proposed a clear design principle for our model: the controllable branch should identify the system dynamics that depend on the agent's actions, the non-controllable branch should capture passive forces surrounding an agent, and the static branch should learn stationary state features. With this design principle, we are able to maintain a unified disentangled representation learning framework as illustrated in Figure 3. It is worth noting that we did not assume $\hat{s}$ and $\hat{v}$ to be mutually independent. In cases where a clear separation is infeasible, we can still disentangle the environment dynamics into modular representations and tackle sub-tasks decoupled from the original complex learning task, using this unified framework.

The world model in our VDFD architecture is highly relevant to the Iso-Dream method (Pan et al., 2022), in which the controllable, non-controllable, and static components are decoupled from visual data to perform representation learning effectively. However, Iso-Dream is essentially the same type of model-based method as Dreamer (Hafner et al., 2020) in that it considers the virtual trajectories generated by the world model as interaction experiences and trains the policy entirely using these trajectories. Moreover, Iso-Dream was applied to visual control tasks, in the same application domain as Dreamer. In contrast, as demonstrated by Figure 3, our method extracts useful knowledge from the imaginary roll-outs as part of the training data for the value decomposition framework, rather than relying only on the imaginations. VDFD is specifically designed for complex MARL tasks, which are fundamentally different from visual control problems.

## 4.2 Combining the Modularized World Model with Value Function Decomposition

We amalgamate the world model with the mixing network in QMIX (Rashid et al., 2018) to make it applicable in multi-agent systems. It operates under the condition of IGM. Receiving outputs of the agent network and merging them monotonically, this mixing network is responsible for the reinforcement learning process in our method. It models the joint action-value function $Q_{tot}(\boldsymbol{\tau}_t, \mathbf{a}_t)$, the only component of the environment that is not included in the disentangled world model. $Q_{tot}(\boldsymbol{\tau}_t, \mathbf{a}_t)$ can be decomposed into individual action-value functions to calculate the expected returns. After combining the value decomposition framework with the world model, we obtain the complete workflow of VDFD. It is displayed in Figure 3. The architecture may seem different from Figure 1b because we use the world model to carry out latent imaginations, where the roll-out horizon for imagination is set to 3.

We will explain how one step forward in the imagination works. At time step $t$, $\mathbf{h}_t$ encapsulates all past information including the joint action-observation histories $\boldsymbol{\tau}_t$. In the action-conditioned module, $\mathbf{h}_t$ is fed

into the agent network. Under the joint policy $\pi = (\pi^1, ..., \pi^N)$, the imagined joint actions can be obtained by $\mathbf{a}'_t \sim \pi(\cdot|\boldsymbol{\tau}_t)$. Then $\mathbf{a}'_t$ is sent by the agent network into the VAE. Meanwhile, the GALA encoder takes $\mathbf{h}_t$ as input and computes the posterior latent state $\hat{s}_t \sim q_\theta(\cdot|\mathbf{h}_t)$, which is also sent into the prior model. With all the required inputs (*i.e.*, $\mathbf{a}'_t$ and $\hat{s}_t$), it can infer the latent state $\hat{s}_{t+1} \sim p_\theta^{\mathrm{prior}}(\cdot|\hat{s}_t, \mathbf{a}'_t)$. Then, $\hat{s}_{t+1}$ is taken by the GALA decoder as input to derive $\mathbf{h}'_{t+1}$, which contains the imagined action-observation histories $\boldsymbol{\tau}'_{t+1}$. The one step forward is completed in this module and the procedures repeat at $t+1$.

Regarding the action-free module, the inputs for GALA at time $t$ are the action-free latent state $\hat{v}_{t-1}$ and joint observations $\mathbf{z}_t$. GALA outputs $\hat{v}'_{t-1}$ and $\hat{\mathbf{z}}_t$ by reconstruction. The prior VAE uses $\hat{v}'_{t-1}$ to deduce $\hat{v}_t \sim p_\theta^{\mathrm{prior}}(\cdot|\hat{v}'_{t-1})$. Then $\hat{v}_t$ is passed into GALA together with $\hat{\mathbf{z}}_t$ for the next step forward. For the static module, we only need to feed the environment observations into GALA.

Suppose the roll-out horizon is set to $j \in \mathbb{Z}$. We perform $j$ roll-outs in the action-conditioned and action-free branches to obtain the latent states $\{\hat{s}_t, ..., \hat{s}_{t+j}\}$ and $\{\hat{v}_t, ..., \hat{v}_{t+j}\}$. These sequences of states are the outputs of the action-conditioned and the action-free branches. They can be expressed as $\mathbf{o}_t^{\mathrm{ac}}$ and $\mathbf{o}_t^{\mathrm{af}}$. They are aggregated to form the roll-out states $\hat{s}_t^{\mathrm{Rollout}}$ and $\hat{v}_t^{\mathrm{Rollout}}$, using real-valued masks $\mathbf{M}^{\mathrm{ac}}$ and $\mathbf{M}^{\mathrm{af}}$. We denote the observations as $\hat{\mathbf{z}}_{1:t+j}$ for the static branch. GALA takes in $\hat{\mathbf{z}}_{1:t+j}$ and returns $\hat{\mathbf{z}}^{\mathrm{static}}$, which is the output of the static branch. It is transformed into $\hat{\mathbf{z}}^{\mathrm{Rollout}}$ through $\mathbf{M}^{\mathrm{st}}$. More specifically, the transformations of the outputs can be written as $\hat{s}_t^{\mathrm{Rollout}} = \mathbf{M}^{\mathrm{ac}} \odot \mathbf{o}^{\mathrm{ac}}$, $\hat{v}_t^{\mathrm{Rollout}} = \mathbf{M}^{\mathrm{af}} \odot \mathbf{o}^{\mathrm{af}}$, and $\hat{\mathbf{z}}^{\mathrm{Rollout}} = \mathbf{M}^{\mathrm{st}} \odot \hat{\mathbf{z}}^{\mathrm{static}}$. It is clear that the transformations are consistent with Equation 3 and the masks are used for the same purpose.

Combining $\hat{s}_t^{\mathrm{Rollout}}$, $\hat{v}_t^{\mathrm{Rollout}}$ and $\hat{\mathbf{z}}^{\mathrm{Rollout}}$ together, we obtain $O_t^{\mathrm{Rollout}}$, building a bridge between world model learning and value decomposition framework, in which the mixing network receives three different types of inputs. The first type consists of individual action-value functions from the agent network, the second is the real global state $s_t$, and the last is $O_t^{\mathrm{Rollout}}$. Because $O_t^{\mathrm{Rollout}}$ incorporates information about potential future observations and states with isolated controllable and non-controllable dynamics, it can assist the agents greatly in decision-making.

### 4.3 Components of the Learning Objective

We divide the optimization of the overall learning objective into two parts. The first part is the decoupled world model, which involves learning simulated data via generative models. The second part is the value decomposition framework, which takes as input the imagined roll-outs and agent networks for the reinforcement learning process.

In the action-conditioned module of the world model, we aim to minimize the KL divergence between the prior and posterior distributions. We need to prevent the prior distribution of $\hat{s}_t$ from becoming extremely complex, meaning that we will consider the KL term between the prior distribution and the reference distribution $p(s_t)$. As discussed in Section 4.1, the actual conditional distribution of the priors, $p(\hat{s}_t|\hat{s}_{t-1}, \mathbf{a}_{t-1})$, is unknown. Inspired by (Huang et al., 2020), we introduce a new parametrized model, $p_\theta^{\mathrm{prior}}(\hat{s}_t|\hat{s}_{t-1}, \mathbf{a}_{t-1})$, to estimate the actual conditional distribution. We construct a VAE for learning $p_\theta^{\mathrm{prior}}(\hat{s}_t|\hat{s}_{t-1}, \mathbf{a}_{t-1})$. In standard VAEs, the prior $p(\cdot)$ can often be set as a standard Gaussian (Huang et al., 2020). Therefore, when we calculate the KL loss for the action-conditioned module, we also need to include the KL divergence between $p_\theta^{\mathrm{prior}}(\hat{s}_t|\hat{s}_{t-1}, \mathbf{a}_{t-1})$ and the standard Gaussian $p(s_t) := \mathcal{N}(0, I)$. Then we obtain:

$$\begin{aligned}
\mathcal{L}_{\mathrm{ac}}^{\mathrm{KL}}(\theta) &= \mathcal{D}_{\mathrm{KL}}\left[p_\theta^{\mathrm{prior}}\left(\hat{s}_t \mid \hat{s}_{t-1}, \mathbf{a}_{t-1}\right) \| p(s_t)\right] + \mathcal{D}_{\mathrm{KL}}\left[q_\theta(\hat{s}_t|\mathbf{h}_t)\|p_\theta^{\mathrm{prior}}\left(\hat{s}_t|\hat{s}_{t-1}, \mathbf{a}_{t-1}\right)\right] \\
&= \mathcal{L}_{\mathrm{ac\text{-}prior}}^{\mathrm{KL}}(\theta) + \mathcal{L}_{\mathrm{ac\text{-}post}}^{\mathrm{KL}}(\theta)
\end{aligned}$$

Reconstruction processes occur in both generative models. The VAE outputs $\hat{s}'_{t-1}$ and $\mathbf{a}'_{t-1}$, which are reconstructed from the original past state and actions, $\hat{s}_{t-1}$ and $\mathbf{a}_{t-1}$. Likewise, GALA outputs $\mathbf{h}'_t$ by reconstructing the authentic past information $\mathbf{h}_t$. The reconstruction loss function can be expressed as:

$$\mathcal{L}_{\mathrm{ac}}^{\mathrm{REC}}(\theta) = \mathcal{L}_{\mathrm{ac\text{-}prior}}^{\mathrm{REC}}(\theta) + \mathcal{L}_{\mathrm{ac\text{-}post}}^{\mathrm{REC}}(\theta)$$

where:

$$\mathcal{L}_{\text{ac-prior}}^{\text{REC}}(\theta) = \text{MSE}(\hat{s}_{t-1}, \hat{s}'_{t-1}; \theta) + \text{MSE}(\mathbf{a}_{t-1}, \mathbf{a}'_{t-1}; \theta)$$
$$\mathcal{L}_{\text{ac-post}}^{\text{REC}}(\theta) = \text{MSE}(\mathbf{h}_{t-1}, \mathbf{h}'_{t-1}; \theta)$$

Here $\text{MSE}(\cdot)$ denotes the mean squared error function. The calculation of loss in the action-free branch is very similar to the action-conditioned branch, except that the auto-encoders receive different inputs and produce different outputs. We can derive the KL divergence loss as:

$$\mathcal{L}_{\text{af}}^{\text{KL}}(\theta) = \mathcal{D}_{\text{KL}} \left[ p_\theta^{\text{prior}} \left( \hat{v}_t \mid \hat{v}_{t-1} \right) \| p(v_t) \right] + \mathcal{D}_{\text{KL}} \left[ q_\theta(\hat{v}_t | \hat{v}_{t-1}, \mathbf{z}_t) \| p_\theta^{\text{prior}} \left( \hat{v}_t | \hat{v}_{t-1} \right) \right]$$
$$= \mathcal{L}_{\text{af-prior}}^{\text{KL}}(\theta) + \mathcal{L}_{\text{af-post}}^{\text{KL}}(\theta)$$

and the reconstruction loss as:

$$\mathcal{L}_{\text{af}}^{\text{REC}}(\theta) = \text{MSE}(\hat{v}_{t-1}, \hat{v}'_{t-1}; \theta) + \text{MSE}((\hat{v}_{t-1}, \mathbf{z}_{t-1}), (\hat{v}'_{t-1}, \mathbf{z}'_{t-1}); \theta)$$
$$= \mathcal{L}_{\text{af-prior}}^{\text{REC}}(\theta) + \mathcal{L}_{\text{af-post}}^{\text{REC}}(\theta)$$

There is no KL term in the static branch of the world model. The loss is computed as:

$$\mathcal{L}_{\text{st}}(\theta) = \text{MSE}(\hat{\mathbf{z}}_{1:t}, \hat{\mathbf{z}}^{\text{static}}; \theta)$$

We implement the KL balancing technique as an option when minimizing the KL loss because we want to avoid the posterior representations being regularized towards a poorly trained prior. To address this problem, we employ different learning rates so that the KL divergence is minimized faster with respect to the prior than the posterior. We apply the $stop\_grad(\cdot)$ function in Hafner et al. (2022), which stops backpropagation from gradients of variables. Let $\alpha \in [0, 1]$ be the learning rate. KL balancing can then be defined as:

$$\mathcal{D}_{\text{KLB}} \left[ q_\theta(\cdot) \| p_\theta^{\text{prior}}(\cdot) \right] = \alpha \mathcal{D}_{\text{KL}} \left[ q_\theta(\cdot) \| stop\_grad(p_\theta^{\text{prior}}(\cdot)) \right] + (1 - \alpha) \mathcal{D}_{\text{KL}} \left[ stop\_grad(q_\theta(\cdot)) \| p_\theta^{\text{prior}}(\cdot) \right]$$

Finally, we denote the learnable parameters of the value decomposition framework as $\phi$. To maintain consistency with QMIX, we define the TD target $y^{tot}$ and TD loss $\mathcal{L}^{\text{TD}}(\phi)$ as:

$$y^{tot} = r_t + \gamma \max_{\mathbf{a}_{t+1}} Q_{tot}(\boldsymbol{\tau}_{t+1}, \mathbf{a}_{t+1}, s_{t+1}, ; \phi)$$
$$\mathcal{L}^{\text{TD}}(\phi) = \left( y^{tot} - Q_{tot}(\boldsymbol{\tau}_t, \mathbf{a}_t, s_t, ; \phi) \right)^2$$

The overall loss function of VDFD can be expressed as:

$$\mathcal{L} = \beta_1 \mathcal{L}_{\text{ac}}^{\text{KL}} + \beta_2 \mathcal{L}_{\text{ac}}^{\text{REC}} + \beta_3 \mathcal{L}_{\text{af}}^{\text{KL}} + \beta_4 \mathcal{L}_{\text{af}}^{\text{REC}} + \beta_5 \mathcal{L}_{\text{st}} + \mathcal{L}^{\text{TD}} \tag{4}$$

where $\{\beta_i\}_{i \in \{1...5\}}$ are hyperparameters corresponding to the real-valued masks[3]. As suggested by Section 4.1.4, if a branch is masked out, the corresponding $\beta_i$ will be set to 0. By jointly optimizing components of the overall learning objective, we guide VDFD to accurately predict latent trajectories and efficiently perform reinforcement learning, eventually maximizing the returns.

## 5    Experiments

To assess the performance and generalization capability of VDFD, we consider three well-established MARL benchmarks: StarCraft Multi-Agent Challenge (SMAC), Multi-Agent MuJoCo (MAMuJoCo), and Level-Based Foraging (LBF). We compare our method with widely applied MARL algorithms, including VDN (Sunehag et al., 2017), IQL (Tampuu et al., 2017), COMA (Foerster et al., 2018), QMIX (Rashid et al.,

---

[3]For choosing the values of $\beta_i$, we had some hyperparameter tuning before presenting the final results in the paper. We have never obtained very poor results, and the reason could be that we did not set $\beta_i$ to be unreasonably high. We report the optimal hyperparameter settings that we have obtained in Appendix C.

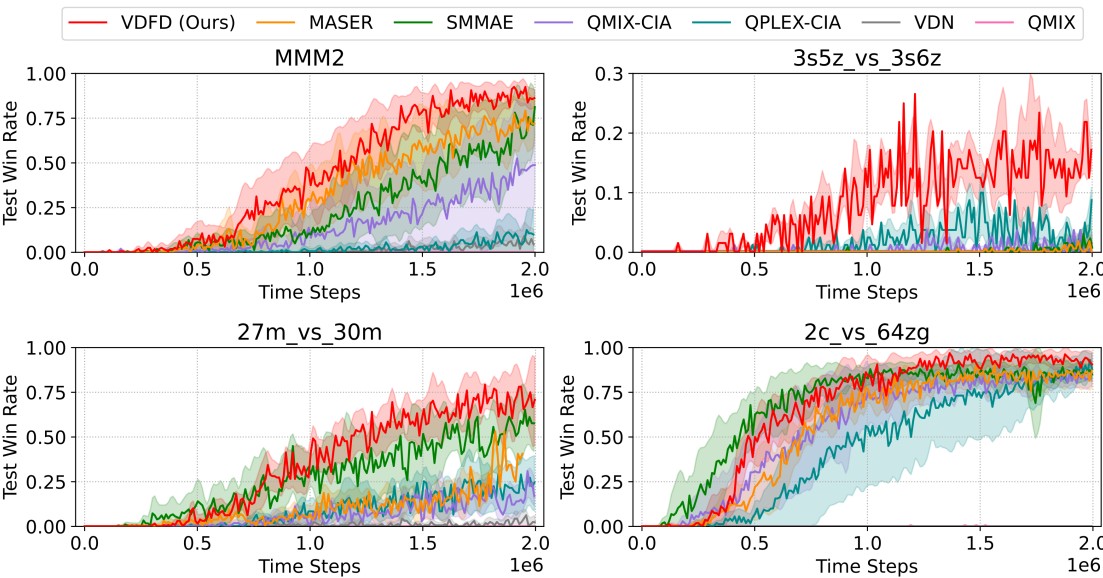

Figure 4: Performance of VDFD against the other algorithms in *Super-Hard* SMAC environments. VDFD yields the best performance overall. SMMAE converges faster on one map (*2c_vs_64zg*) but our method surpasses the win rate of SMMAE by 7.96% ultimately. Please refer to Appendix B for all nine baselines implemented and the results on the other SMAC battle scenarios.

2018), QTRAN (Son et al., 2019), MASER (Jeon et al., 2022), QMIX-CIA (Liu et al., 2023), SMMAE (Zhang et al., 2023), and QPLEX-CIA (Liu et al., 2023). We make sure the baselines for our experiments include both the well-known methods that demonstrate the CTDE paradigm (e.g., VDN, QMIX, QTRAN) and the MARL algorithms that are the most up-to-date (e.g., MASER, QMIX-CIA, SMMAE). All experiments are performed with five different seeds. Moreover, we carry out ablation studies to analyze the impact of the main components of VDFD, involving different branches in the disentangled world model, the variational graph auto-encoder, and the approximate prior function. We verify that they are indispensable to our method.

### 5.1 Performance on StarCraft II Micro-Management

Built upon StarCraft II, SMAC consists of battle scenarios corresponding to an extensive range of learning tasks (Samvelyan et al., 2019). It focuses on the problem of micro-management, in which each agent takes fine-grained control over an individual unit and selects actions independently. As SMAC requires the official API, we use the newest SC2.4.10 release (Blizzard, 2019). It contains stability updates and bug fixes compared to the older releases adopted in previous papers (Rashid et al., 2018; Mahajan et al., 2019; Du et al., 2019; Jeon et al., 2022). Every agent manipulating a unit receives observations only within the unit's field of view, which leads to partial observability. The SMAC scenarios (maps) can be classified into three categories: *Easy*, *Hard*, and *Super-Hard*, based on the difficulty. We evaluate the VDFD model on 21 diverse scenarios, all of which can be found in Appendix B. All experiments in SMAC run for 2M time steps. We plot the mean and the standard deviation of the five independent runs for each algorithm. The main results are summarized as *Test Win Rate*, defined as the percentage of episodes in which our army defeats the enemies within the permitted time limit (Samvelyan et al., 2019). Table 1 presents an overview of the main results for VDFD and the other four competing methods in the SMAC benchmark.

Figure 4 indicates that VDFD rises above the competing methods on the *Super-Hard* SMAC challenges. Compared to *Easy* and *Hard* scenarios, the *Super-Hard* challenges require more effective cooperation and particular micro-management tricks to defeat opponents. In *3s5z_vs_3s6z*, the enemies have one more Zealot than the allies, which enables them to isolate the ally Zealots from the ally Stalkers and attack the latter. *MMM2* is asymmetric and heterogeneous, in which there are three different types of units on each side. The enemy army in *2c_vs_64zg* consists of 64 Zerglings, yielding the largest state and action spaces

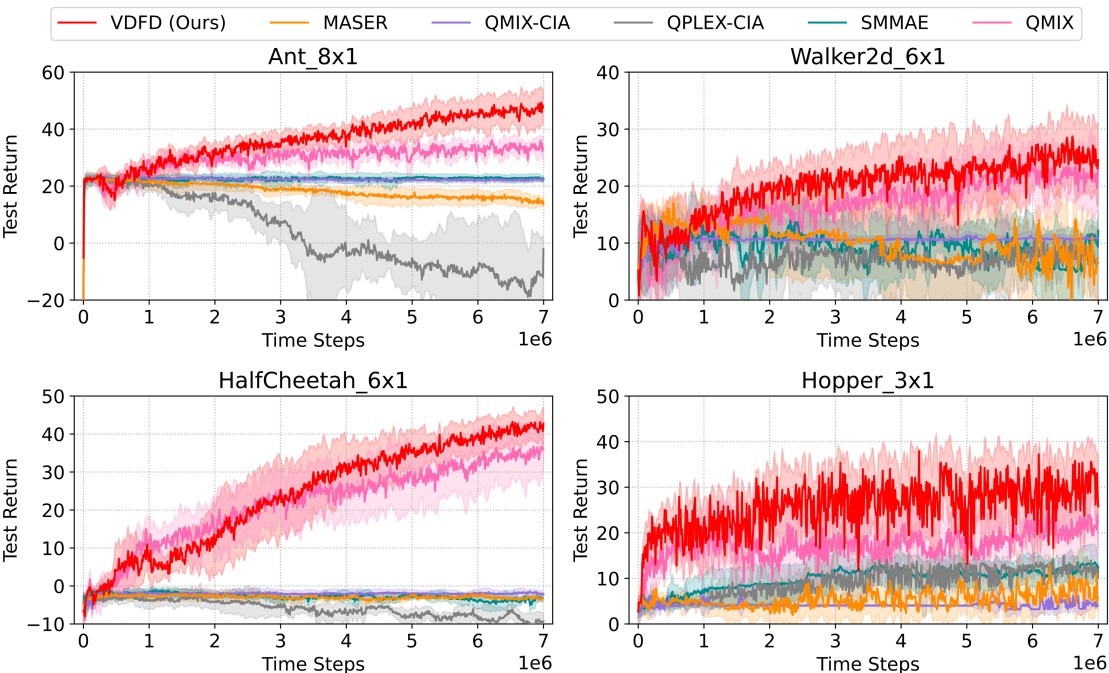

Figure 5: Episodic return of VDFD compared to other methods in Multi-Agent MuJoCo control suite. The return is scaled for clear plotting. Appendix C contains experimental details. Each run lasts 7M time steps. VDFD is able to gain the highest return during the testing phase in every scenario. Our algorithm outperforms QMIX, which shows the highest return among the baselines, by 25% on average.

in SMAC. It is followed by *27m_vs_30m*, which consists of 57 Marines in total. Because VDFD leverages the decoupled world model that performs latent imaginations, it is equipped with useful knowledge about what will happen in the future given different actions. It can therefore maintain precise control over the ally units and adopt the optimal micro-management strategy for a certain environment more easily than other baselines. Furthermore, since VDFD is essentially a model-based approach, it can be more sample-efficient than model-free methods, especially in large-scale battles.

## 5.2 Performance on Multi-Agent MuJoCo

To address the need for multi-agent robotic control, Peng et al. (2021) developed Multi-Agent MuJoCo (MAMuJoCo) based on the original MuJoCo suite Todorov et al. (2012). In MAMuJoCo, a single robot can be partitioned into disjoint sub-graphs, each of which represents an agent. The agent may control one or more joints, depending on the specific agent partitioning. All agents need to collaborate on solving diverse tasks. MAMuJoCo also introduces partial observability, as the agents can be configured with different levels of observational capabilities. For example, an agent can be set to observe only the state of its own joints and body, or it can be set to observe its immediate neighbor's joints and bodies. In our experimental settings, each of the agents is constrained to observe only the two nearest joints. An overview of the results on MAMuJoCo is presented in Table 2.

As Figure 5 shows, four different learning tasks are completed[4]. Several methods face difficulties in performing MAMuJoCo tasks. For example, QPLEX-CIA and MASER even obtain negative returns in one or more scenarios, because the robot cannot move forward and consequently gets penalized. In contrast, VDFD gains the highest episodic return in every environment, demonstrating that it consistently learns to move

---

[4] *Ant_8×1* is the Ant partitioned into 8 agents, *Walker2d_6×1* is the Walker partitioned into 6 agents, *HalfCheetah_6×1* is the Half Cheetah partitioned into 6 agents, and *Hopper_3×1* is the Hopper partitioned into 3 agents. All of the tasks are partially observable.

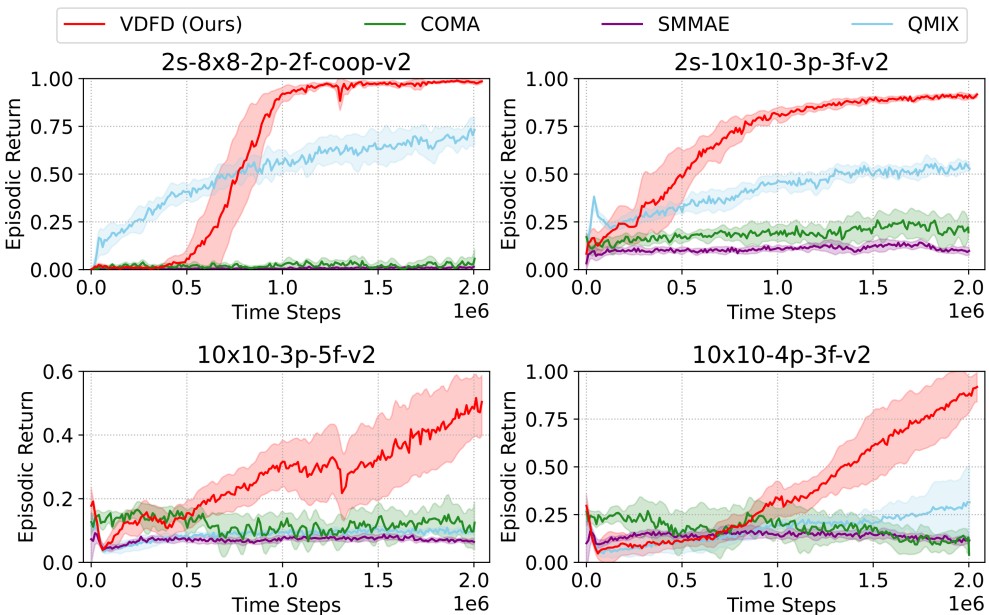

Figure 6: The episodic return of VDFD compared with the baselines in the benchmark of Level-Based Foraging. We select four representative tasks in which the levels of partial observability and cooperation vary, and in which the numbers of players and food items differ. The outstanding competence of VDFD is evident in all of the tasks. Our method exceeds QMIX, the second-best competitor, by a large margin.

the robots in the correct direction. Because VDFD leverages simulated roll-outs in policy improvement, it is able to learn policies that coordinate the distinct joints of the robot under different configurations.

### 5.3 Performance on Level-Based Foraging

The Level-Based Foraging environment (LBF) Christianos et al. (2020) is a collection of mixed cooperative-competitive games in the multi-agent domain. The general setting consists of agents and food items, which are placed in a grid world and are assigned different levels. The goal of agents is to collect food, but the collection of a food item is allowed only if the sum of levels of agents involved in loading is greater or equal to the item's level. Agents receive a reward equal to the level of the collected food divided by their contribution (i.e., their levels). LBF poses challenges to both the cooperation and the competition of the agents. In this paper, four distinct LBF tasks are defined, with variable numbers of agents and items, cooperation setting, partial observability, and world size. Detailed settings regarding the LBF, the SMAC, and the MAMuJoCo benchmarks can be found in Appendix A and Appendix C. Table 4 provides an overview of our results across these three benchmark environments.

In Figure 6 and Table 3, it is clear that VDFD surpasses all other baselines in collecting food items and maximizing episodic returns. QMIX exhibits moderate performance when the world is relatively small (in "*2s-8x8-2p-2f-coop-v2*"), but degrades significantly as the world size, the number of players, or the number of food items grows. In the task of "*10x10-3p-5f-v2*", there are three players with more items to collect (five items) than in other worlds. The players, therefore, need to spend more time gaining rewards. However, VDFD still achieves remarkable progress given the limited time steps.

### 5.4 Ablation Studies

We carry out ablation experiments to investigate the effects of the main components in VDFD. First of all, the performance comparison between VDFD and QMIX is implicitly an ablation study that shows the importance of combining the world model with value function factorization.

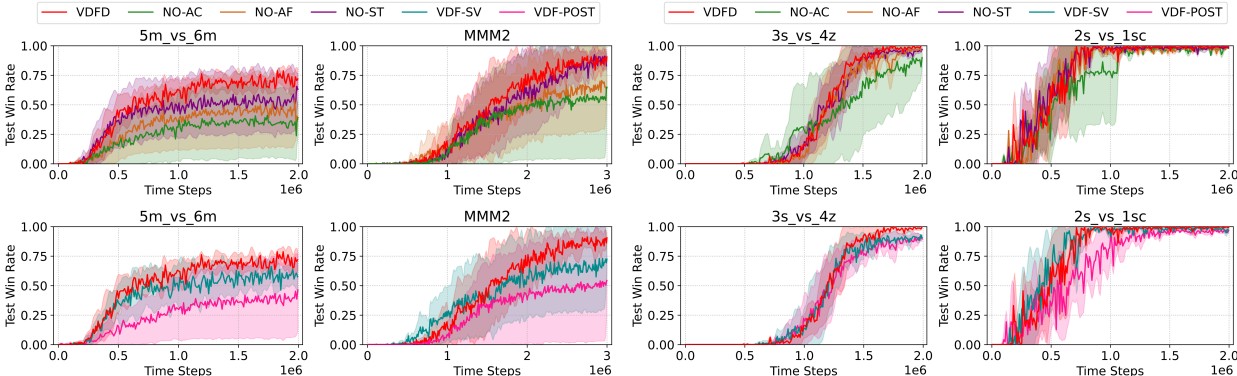

Figure 7: Comparison of VDFD win rates with two different sets of ablations on four SMAC scenarios. The two sets of ablations are {NO-AC, NO-AF, NO-ST} and {VDF-SV, VDF-POST}. The SMAC environments are *5m_vs_6m*, *MMM2*, *3s_vs_4z* and *2s_vs_1sc*. Each plot displays the mean and standard deviation over five seeds. VDFD outperforms all the ablations, although the difference in win rates is small on the relatively simpler *2s_vs_1sc* map. The results further suggest the importance of disentangled representation learning, prior estimation with VAE, and posterior estimation with GALA in our model-based approach.

Secondly, to examine the influence of the modules in the disentangled world model, we perform a set of experiments with three alterations of VDFD: NO-AC, NO-AF, and NO-ST. In Figure 7, we present the ablation results of them. For the first alteration, NO-AC, we remove the action-conditioned branch of the world model by masking out the outputs of the branch. Likewise, NO-AF and NO-ST are implemented to mask out action-free and static branches. It is clear that the lack of any module in the world model results in sub-optimal performance, as they correspond to different parts of the information about the environment dynamics. NO-AC suffers the most because the agent network and the action representations are excluded from the imagination process. As the complexity of the environment increases, the significance of the action-free branch becomes evident. The performance of the NO-AF model drops 33% in *5m_vs_6m* and 28% in *MMM2*, which is substantial compared to the relatively easier *2s_vs_1sc* map.

Finally, we conduct an ablation study on the impact of generative models. We consider two variants of VDFD in this ablation study, VDF-SV and VDF-POST. The graph auto-encoder GALA is substituted with a fully-connected VAE for learning the posterior $q_\theta(\cdot)$ in VDF-SV. For VDF-POST, we exclude the VAE for learning $p_\theta^{\text{prior}}(\cdot)$, which means that only the posterior GALA is used for state inference. Figure 7 reveals that both VDF-SV and VDF-POST are outclassed by VDFD. VDF-POST has the worst win rates in every scenario. Compared to the original VDFD, the performance of VDF-SV decreases by 19%, 18%, and 11% in *5m_vs_6m*, *MMM2*, and *3s_vs_4z*, respectively. This indicates that GALA excels fully-connected VAE in seizing the information about the relationship between agents. The results of VDF-POST imply that it is necessary to approximate the unknown prior, instead of using only the posterior in representation learning.

## 6 Conclusion

We present VDFD, a model-based method coalescing disentangled representation learning into value function factorization, to address the immense complexity of MARL. To optimize the learning objective, we model the system as Dec-POMDP and integrate generative models in VDFD. We encapsulate the environment representations into a decoupled world model to enhance sample efficiency. VDFD acquires the capability of inferring future states by learning the latent trajectories in the world model. We demonstrate experimentally and analytically that VDFD outstrips well-known MARL baselines and achieves solid generalizability. Using ablation studies, we confirm that the components of VDFD are indispensable for its outstanding performance.

Looking ahead, we aim to implement different generative models, such as new variants of the VAE, to compare their effectiveness when applied to the world model. Since the experiments are conducted in MARL environments rather than with image datasets, it can be very difficult to obtain intuitive visualizations of

Table 1: An overview on the Test Win Rates of VDFD compared with MASER, SMMAE, QMIX-CIA, and QPLEX-CIA in *Super-Hard*, *Hard*, and *Easy* SMAC environments. Even in the challenge of *2c_vs_64zg* where the competing algorithms display the smallest performance differences among *Super-Hard* environments, our method is still on top of the others at the end of the testing episodes, with an average improvement of 6.42%. The baselines that completely fail in the *Super-Hard* challenges (such as COMA and IQL) are not considered for comparison.

| SMAC Environments\Methods | VDFD | MASER | SMMAE | QMIX-CIA | QPLEX-CIA |
|---|---|---|---|---|---|
| *2c_vs_64zg* | **0.92** | 0.87 | 0.85 | 0.85 | 0.88 |
| *3s5z_vs_3s6z* | **0.18** | 0.02 | 0.01 | 0.02 | 0.10 |
| *27m_vs_30m* | **0.71** | 0.41 | 0.58 | 0.17 | 0.25 |
| *MMM2* | **0.86** | 0.72 | 0.81 | 0.49 | 0.11 |
| | | | | | |
| SMAC *Easy* Challenges (Averaged) | 0.98 | **0.99** | 0.95 | 0.97 | 0.98 |
| SMAC *Hard* Challenges (Averaged) | **0.95** | 0.87 | 0.86 | 0.85 | 0.87 |

Table 2: An overview on the episode return of VDFD compared with other MARL baselines in Multi-Agent MuJoCo. It is clear that among all the competing baselines, QMIX and VDN acquire the highest episodic return at the end of the testing episodes. VDFD surpasses QMIX by 25% and VDN by 61% across different Multi-Agent MuJoCo environments on average.

| Tasks | VDFD | MASER | IQL | QMIX-CIA | QPLEX-CIA | SMMAE | QMIX | VDN |
|---|---|---|---|---|---|---|---|---|
| *Ant_8×1* | **48.54** | 13.65 | 22.50 | 22.23 | -2.09 | 22.62 | 33.50 | 26.73 |
| *Walker2d_6×1* | **24.59** | 7.40 | 18.61 | 10.86 | 6.56 | 12.16 | 23.33 | 18.72 |
| *HalfCheetah_6×1* | **42.73** | -3.14 | 16.03 | -2.18 | -9.26 | -3.28 | 35.83 | 20.13 |
| *Hopper_3×1* | **28.74** | 5.86 | 17.09 | 3.94 | 12.18 | 12.52 | 22.09 | 24.10 |

Table 3: An overview on the episode return of VDFD compared to the other approaches in Level-Based Foraging. Our method can make substantial performance gains across distinct LBF tasks compared to QMIX, VDN, and IQL.

| LBF Environments\Methods | VDFD | COMA | SMMAE | QMIX | VDN | IQL |
|---|---|---|---|---|---|---|
| *2s-8x8-2p-2f-coop-v2* | **0.99** | 0.06 | 0.02 | 0.73 | 0.71 | 0.65 |
| *2s-10x10-3p-3f-v2* | **0.92** | 0.20 | 0.10 | 0.53 | 0.56 | 0.56 |
| *10x10-3p-5f-v2* | **0.50** | 0.12 | 0.07 | 0.11 | 0.10 | 0.11 |
| *10x10-4p-3f-v2* | **0.92** | 0.04 | 0.12 | 0.31 | 0.57 | 0.37 |

Table 4: An overview of the performance of VDFD compared to competing methods across SMAC, MA-MuJoCo, and LBF. For SMAC, Test Win Rates are averaged within difficulty levels. For MAMuJoCo and LBF, the episodic returns are averaged over all tasks selected within the benchmark. VDFD leads the group among all environments, except for the *Easy* SMAC maps where results are nearly identical.

| Environments\Methods | VDFD | MASER | SMMAE | QMIX-CIA | QPLEX-CIA | QMIX |
|---|---|---|---|---|---|---|
| SMAC *Easy* Maps (Avg.) | 0.98 | **0.99** | 0.95 | 0.97 | 0.98 | 0.96 |
| SMAC *Hard* Maps (Avg.) | **0.95** | 0.87 | 0.86 | 0.85 | 0.87 | 0.14 |
| SMAC *Super-Hard* Maps (Avg.) | **0.67** | 0.50 | 0.56 | 0.38 | 0.33 | 0.00 |
| MAMuJoCo Tasks (Avg.) | **36.15** | 5.94 | 11.01 | 8.71 | 1.85 | 28.69 |
| LBF Environments (Avg.) | **0.83** | N/A | 0.08 | N/A | N/A | 0.42 |

disentangled representation learning. Therefore, we plan to adopt the VDFD framework for multi-agent visual control tasks and use images as input to the world model in our future work. Moreover, we are committed to extending the scope of our research to encompass challenging sparse-reward environments. These settings pose a difficult hurdle in MARL, and addressing these challenges holds immense significance for real-world applications. Through comprehensive assessments in such environments, we hope to uncover the potential applicability of VDFD across a broader spectrum of practical domains.

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

# Appendix

## A    Overview of the Three Benchmark Environments

### A.1    StarCraft Multi-Agent Challenge

Our algorithm is tested in the StarCraft Multi-Agent Challenge (SMAC) (Samvelyan et al., 2019). Designed using the popular RTS game StarCraft II, SMAC fosters the development of MARL methods in complex and real-time settings. It encourages the performance analysis of diverse MARL algorithms on standardized benchmarks.

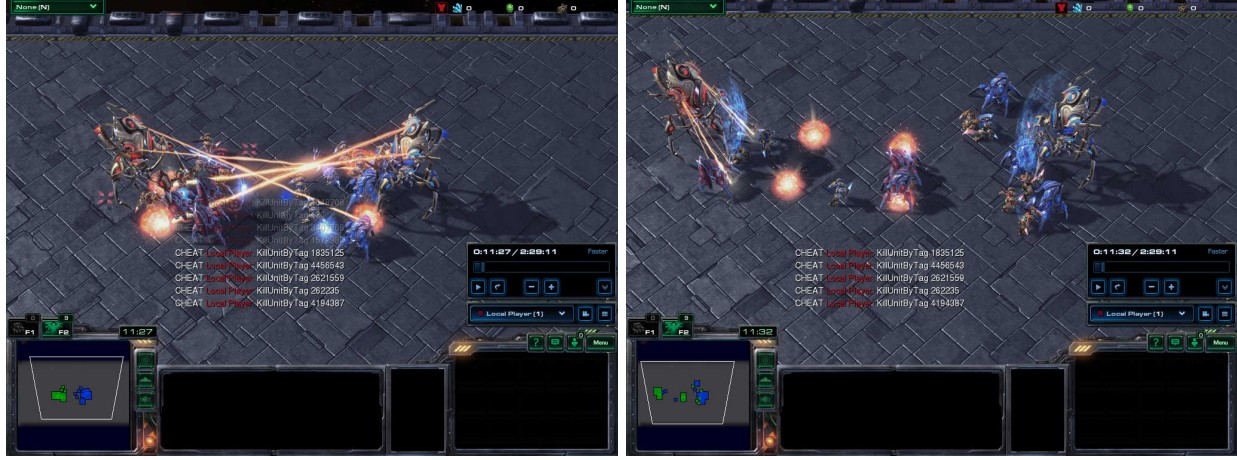

Figure 8: The battle scenario of *1c3s5z* as viewed in the StarCraft II official API. The allied units are set to red and the enemy units are set to blue. Each side has 1 Colossus, 3 Stalkers, and 5 Zealots. This scenario is symmetric but heterogeneous. The allied army has the same number of units as the enemy army (nine units in total), but the team consists of different types of units (three types in total).

A variety of micro-management tasks that SMAC provides are shown in Table 5. They vary in terms of the number and types of units, creating a diverse set of challenges that require effective cooperation and strategies. For example, although *bane_vs_bane* is symmetric, it has 24 units in one team, and 4 of them are Banelings. They are suicide bomber units that explode when being killed, which increases the difficulty of battles. The 7 allied Zealots in *so_many_baneling* have to survive the attacks by 32 enemy Banelings that are strong against Zealots. They must design a plan for positioning, spreading far from each other

| Name | Ally Units | Enemy Units | Type | Difficulty |
|------|-----------|-------------|------|------------|
| 1c3s5z | 1 Colossus, 3 Stalkers & 5 Zealots | 1 Colossus, 3 Stalkers & 5 Zealots | Heterogeneous & Symmetric | Easy |
| 2c_vs_64zg | 2 Colossi | 64 Zerglings | micro-trick: positioning | Super-Hard |
| 2m_vs_1z | 2 Marines | 1 Zealot | micro-trick: alternating fire | Hard |
| 2s_vs_1sc | 2 Stalkers | 1 Spine Crawler | micro-trick: alternating fire | Hard |
| 2s3z | 2 Stalkers & 3 Zealots | 2 Stalkers & 3 Zealots | Heterogeneous & Symmetric | Easy |
| 3m | 3 Marines | 3 Marines | Homogeneous & Symmetric | Easy |
| 3s_vs_3z | 3 Stalkers | 3 Zealots | micro-trick: kiting | Hard |
| 3s_vs_4z | 3 Stalkers | 4 Zealots | micro-trick: kiting | Hard |
| 3s_vs_5z | 3 Stalkers | 5 Zealots | micro-trick: kiting | Hard |
| 3s5z | 3 Stalkers & 5 Zealots | 3 Stalkers & 5 Zealots | Heterogeneous & Symmetric | Hard |
| 3s5z_vs_3s6z | 3 Stalkers & 5 Zealots | 3 Stalkers & 6 Zealots | Heterogeneous & Asymmetric | Super-Hard |
| 5m_vs_6m | 5 Marines | 6 Marines | Homogeneous & Asymmetric | Hard |
| 8m | 8 Marines | 8 Marines | Homogeneous & Symmetric | Easy |
| 8m_vs_9m | 8 Marines | 9 Marines | Homogeneous & Asymmetric | Hard |
| 10m_vs_11m | 10 Marines | 11 Marines | Homogeneous & Asymmetric | Hard |
| 25m | 25 Marines | 25 Marines | Homogeneous & Symmetric | Hard |
| 27m_vs_30m | 27 Marines | 30 Marines | Homogeneous & Asymmetric | Super-Hard |
| bane_vs_bane | 20 Zerglings & 4 Banelings | 20 Zerglings & 4 Banelings | micro-trick: positioning | Hard |
| MMM | 1 Medivac, 2 Marauders & 7 Marines | 1 Medivac, 2 Marauders & 7 Marines | Heterogeneous & Symmetric | Hard |
| MMM2 | 7 Marines, 2 Marauders & 1 Medivac | 8 Marines, 3 Marauders & 1 Medivac | Heterogeneous & Asymmetric | Super-Hard |
| so_many_banelings | 7 Zealots | 32 Banelings | micro-trick: positioning | Hard |

Table 5: StarCraft II micro-management challenges.

on the terrain so that the Banelings' suicidal attacks inflict minimal damage on them. The allied army in *25m* has 25 Marines, far exceeding *3m* and *8m*. Asymmetric scenarios, such as *so_many_baneling* and *2m_vs_1z*, require the agents' effective control over the units to beat the enemies consistently. In *3s_vs_4z* and *3s_vs_5z*, the allied Stalkers need a specific strategy called kiting in order to vanquish the enemy Zealots, which causes delays to the reward.

In Appendix B, we will use a larger number of images similar to Figure 8 to illustrate the behaviors of agents during StarCraft II battles.

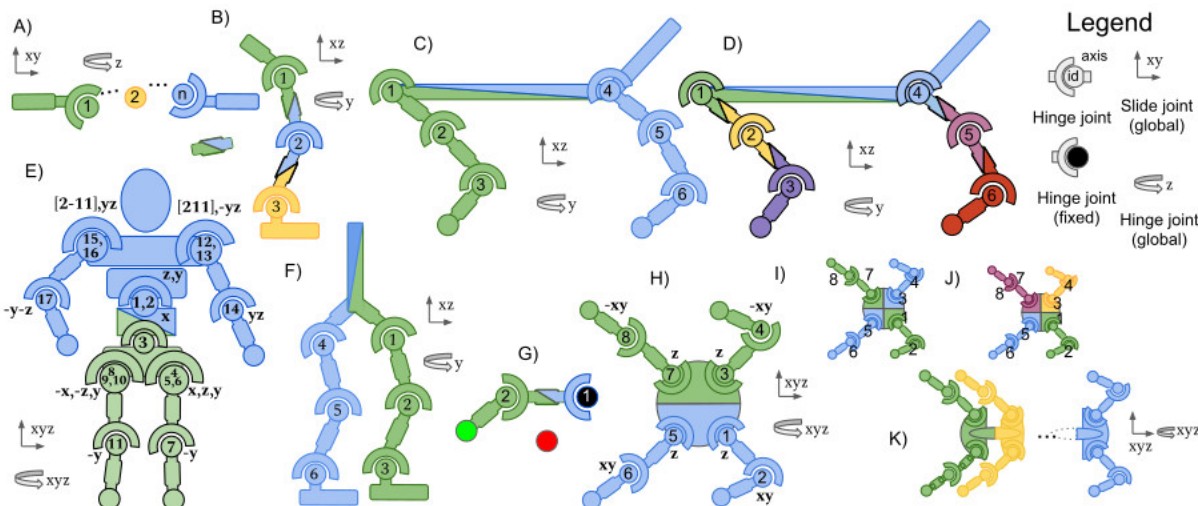

Figure 9: Different configurations of robots in Multi-Agent MuJoCo. A) Many-Agent Swimmer; B) Three-Agent Hopper; C) Two-Agent Half Cheetah; D) Six-Agent Half Cheetah; E) Two-Agent Humanoid; F) Two-Agent Walker2d; G) Two-Agent Reacher; H) Two-Agent Ant; I) Diagonal Two-Agent Ant; J) Four-Agent Ant; K) Many-Agent Ant. The figure is extracted from (Peng et al., 2021, p. 12214).

## A.2   Multi-Agent MuJoCo

The Multi-Agent MuJoCo (MAMuJoCo) environment, developed by Peng et al. (2021), is a benchmark for multi-agent robotic control and learning. It is based on OpenAI MuJoCo environments, which have been widely applied in the research area of single-agent RL. As demonstrated by Figure 9 (Peng et al., 2021, p. 12214), a single robot in the MAMuJoCo environment can be partitioned into disjoint sub-graphs, each of which represents an agent. The agent may control one or more joints, depending on the specific agent partitioning. Each of the joints corresponds to a controllable motor. All agents need to collaborate on solving distinct tasks. In this environment, agents can be configured with different levels of observational capabilities. For instance, an agent can be set to observe only the state of its own joints and body, or it can be set to observe its immediate neighbor's joints and bodies.

The MAMuJoCo configurations used in this paper include:

- The Three-Agent partitioning of Hopper denoted as *Hopper_*3×1 (configuration B in Figure 9),

- The Six-Agent partitioning of Half Cheetah denoted as *HalfCheetah_*6×1 (configuration D in Figure 9),

- The Six-Agent partitioning of Walker2d denoted as *Walker2d_*6×1 (every agent controls a joint of Walker2d),

- The Eight-Agent partitioning of Ant denoted as *Ant_*8×1 (every agent controls a joint of Ant).

## A.3   Level-Based Foraging

The Level-Based Foraging (LBF) environment (Christianos et al., 2020; Papoudakis et al., 2020) is a collection of mixed cooperative-competitive games in the field of multi-agent reinforcement learning. In this environment, agents and food items are randomly placed in a grid world. Each of them is assigned a level, and the level may vary. The task for each agent is to navigate the grid world map and collect the food items. In order to load the items, agents have to choose a certain action next to the item. However, such collection is only successful if the sum of levels of agents involved in loading food is equal to or greater than the food item level. Agents receive a reward equal to the level of the collected food divided by their contribution

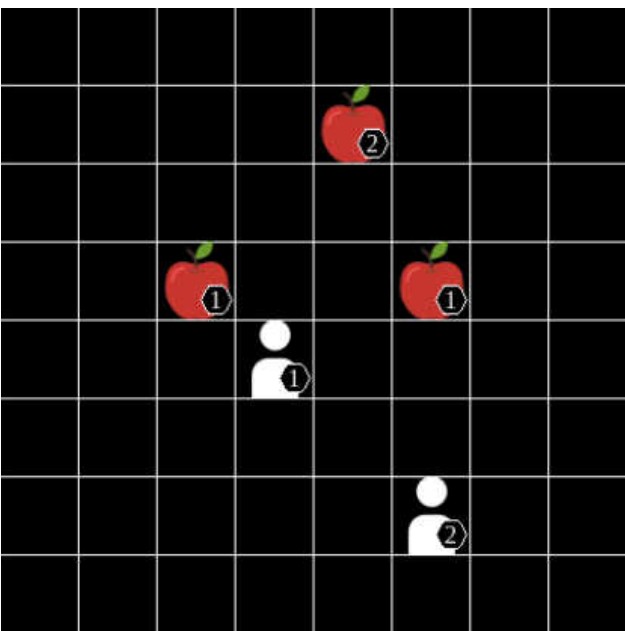

Figure 10: Visualization of a sample task in the Level-Based Foraging environment. The figure is extracted from (Papoudakis et al., 2020, p. 17). This grid world has a size of $8 \times 8$. Two agents collaborate on collecting the three food items randomly located in the grid world.

(their levels). LBF provides a test bed for studying both cooperation and competition among agents in MARL. In our experiments, four distinct LBF tasks are defined, with variable environment configurations.

To customize the environment configuration, Christianos et al. (2020) provide a template: "Foraging{obs}-{x_size}x{y_size}-{n_agents}p-{food}f{force_c}-v2". The options in the template are as follows:

- {obs}: This option introduces partial observability into the environment. For example, if it is set to "-2s", each agent will only have a visibility radius of 2. Otherwise, it can be left blank.

- {x_size} and {y_size}: These fields determine the size of the grid world. For example, an $8 \times 8$ grid world is expressed as "8x8" in the template.

- {n_agents}: This option sets the number of agents (players) for the task.

- {food}: This field sets the number of food items that will be randomly placed in the grid world.

- {force_c}: This field indicates whether the task will be fully cooperative or not. If it is set to "-coop", then the environment will only contain food items that require the cooperation of all agents to be collected successfully.

For example, we have created an LBF environment named "*2s-8x8-2p-2f-coop-v2*" for the experiments. This configuration indicates that there are two players and two food items randomly scattered in an $8 \times 8$ grid world. Every player has a visibility radius of two and they have to fully collaborate on collecting the food.

# B Supplemental Results

## B.1 Test Win Rates of All Methods in SMAC Scenarios

For each SMAC scenario, we will present two plots. This is because we have implemented ten algorithms in total: VDFD (our method), IQL Tampuu et al. (2017), VDN Sunehag et al. (2017), COMA Foerster et al. (2018), QMIX Rashid et al. (2018), QTRAN Son et al. (2019), MASER Jeon et al. (2022), QMIX-CIA Liu et al. (2023), SMMAE Zhang et al. (2023), and QPLEX-CIA Liu et al. (2023). We need to divide them into two groups when we are plotting so that every curve can be viewed clearly.

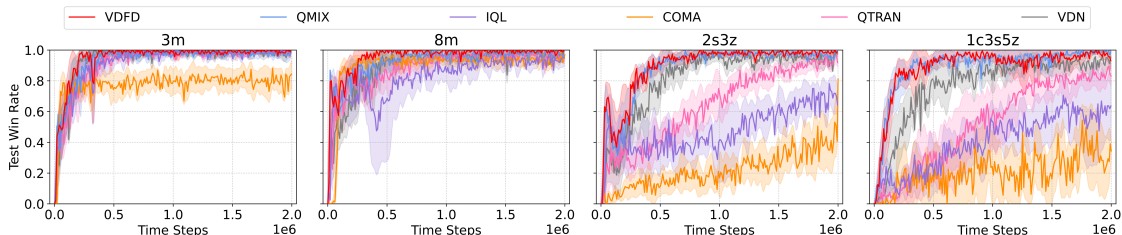

Figure 11: Test Win Rates of VDFD against the baselines of QMIX, IQL, COMA, QTRAN, and VDN in *Easy* SMAC environments.

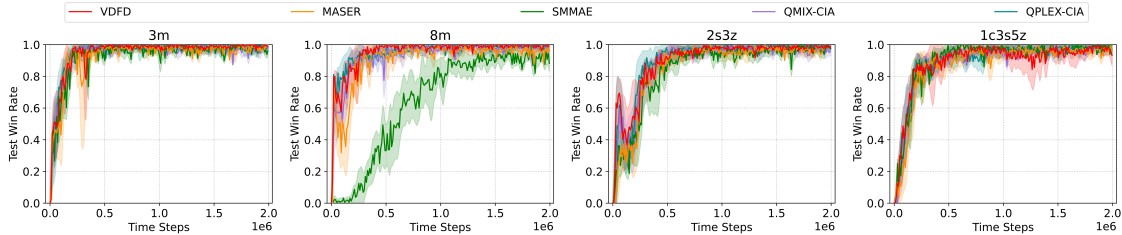

Figure 12: Test Win Rates of VDFD against MASER, SMMAE, QMIX-CIA, and QPLEX-CIA in *Easy* SMAC environments. Every algorithm in this figure exhibits state-of-the-art performance on *Easy* maps.

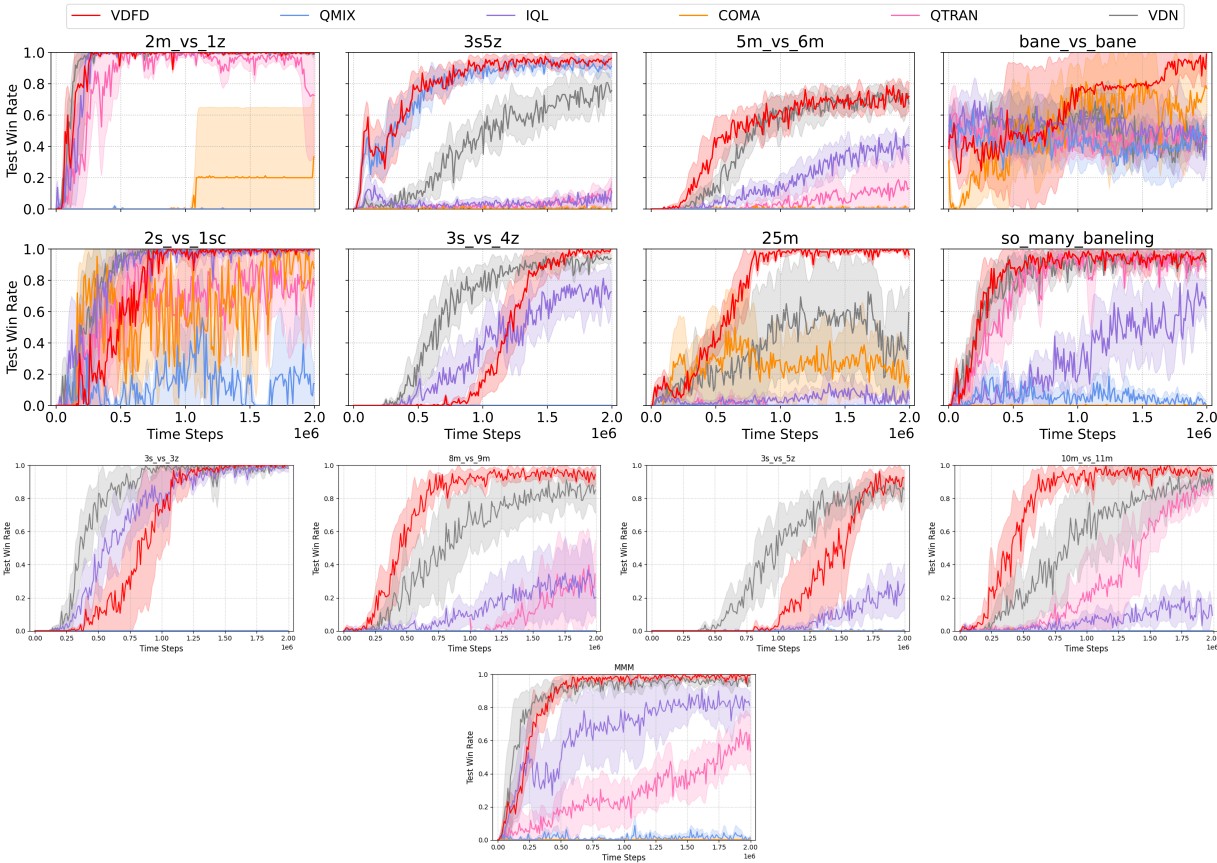

Figure 13: Test win rates of VDFD compared to the baselines of QMIX, IQL, COMA, QTRAN, and VDN in *Hard* challenges. Each plot displays the mean and standard deviation over 5 seeds. There exists a performance gap between *Easy* and *Hard* challenges for the baselines. Neither QMIX nor QTRAN yields high winnnig rates in most of the *Hard* maps. It is obvious that VDFD reaches state-of-the-art performance in all of the additional challenges. VDFD exhibits a delay in time for winning against the enemy army in *3s_vs_5z* since it slowly learns the *kiting* strategy, which requires luring the enemies while maintaining a safe distance.

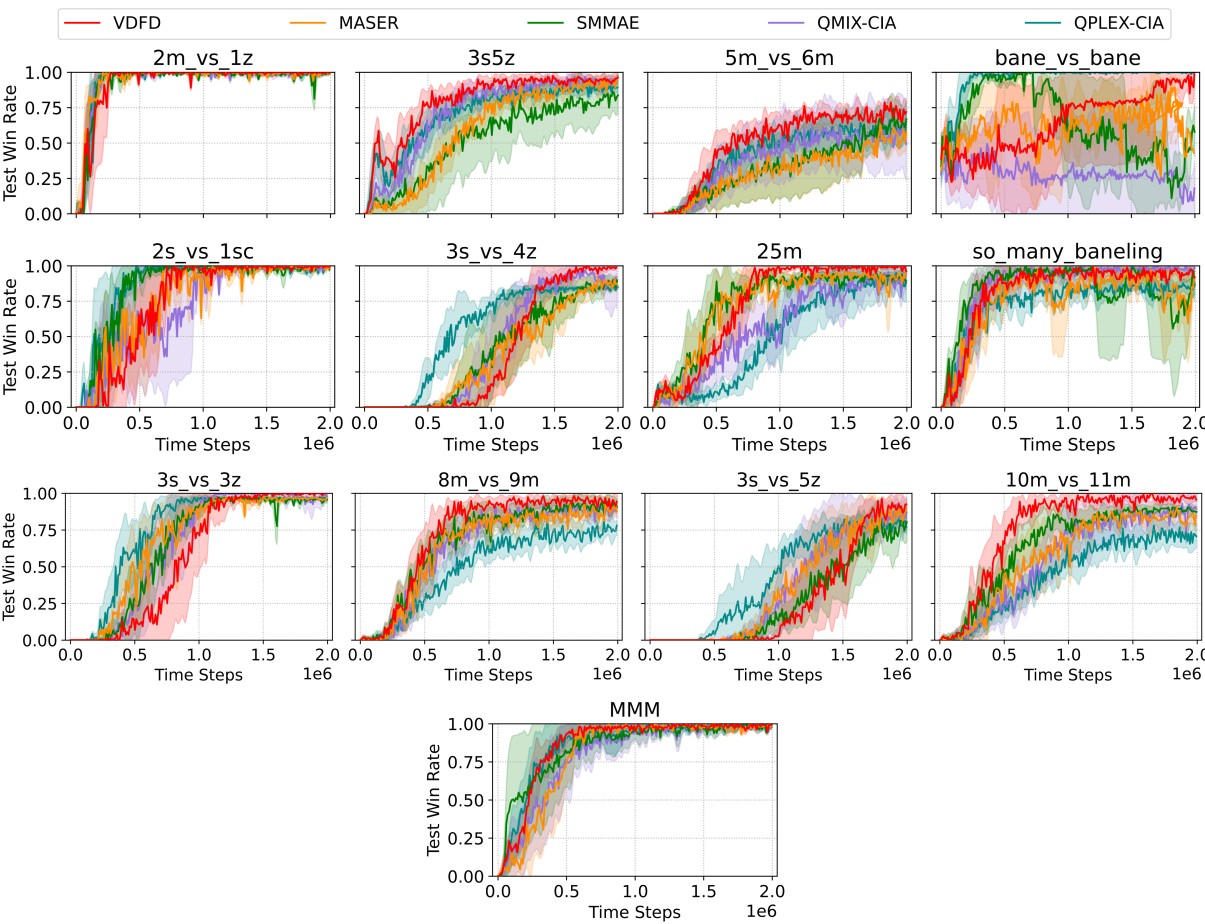

Figure 14: Test win rates of VDFD compared with latest MARL algorithms including SMMAE, MASER, QMIX-CIA, and QPLEX-CIA in *Hard* challenges. Each plot displays the mean and standard deviation over 5 seeds. As newly developed MARL approaches, SMMAE, MASER, QMIX-CIA, and QPLEX-CIA outshine the relatively older baselines shown in the last figure, reaching the same level of competence as VDFD in *Hard* scenarios. Meanwhile, VDFD is still able to surpass all other approaches on some of the maps, such as *5m_vs_6m*, *10m_vs_11m*, and *25m*.

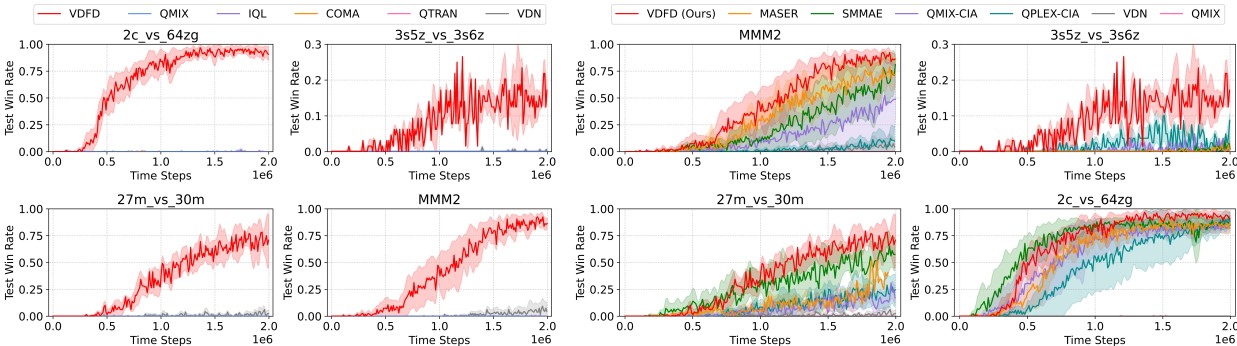

Figure 15: Test win rates of VDFD compared to the other MARL approaches in *Super-Hard* challenges. The plots on the left indicate that the baselines can hardly learn a policy to beat the enemy troops throughout the battle. On the other hand, our method is able to lead the group in terms of performance amongst all the newly proposed algorithms, as demonstrated by the plots on the right.

## B.2 Case Studies of the VDFD Model in StarCraft II Gameplay

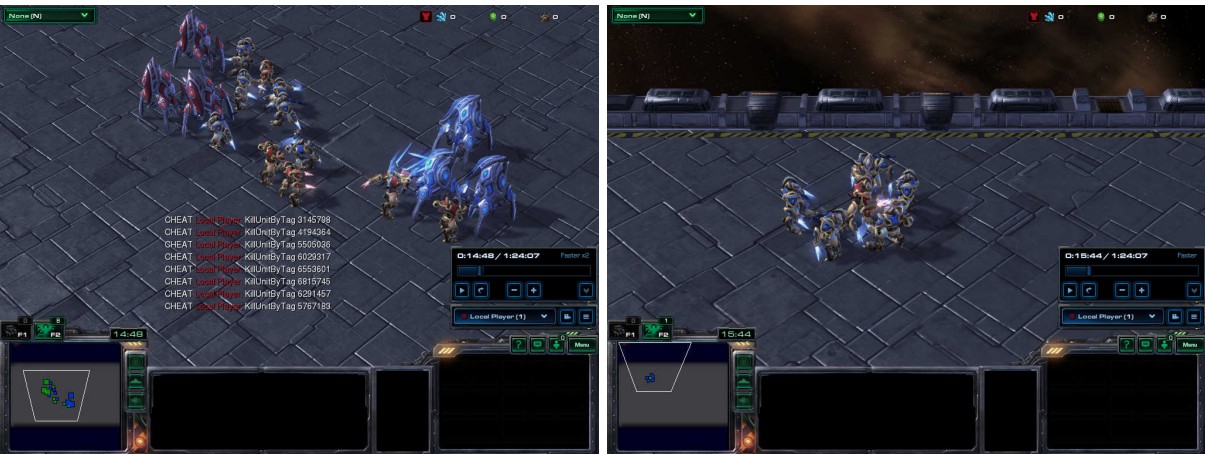

Figure 16: Our model VDFD in *3s5z_vs_3s6z*.

Figure 17: The VDN baseline in *3s5z_vs_3s6z*.

Using the visualization of the experiments in SMAC, here we provide more fascinating details obtained from the official StarCraft II game API. The figures in this section will demonstrate the interactions of the agents during the battles as well as the performance of our disentangled model. Based on the default game setting, the red units belong to the allied army while the blue units represent the enemy troops. Moreover, in the

bottom left corner of each figure, an overview of the map is displayed, in which the green squares and the blue squares denote the allied units and the enemy units, respectively.

Figures 16 and 18 present the agents controlled by VDFD in the *3s5z_vs_3s6z* and *MMM2* environments, respectively. Both of them belong to the classification of *Super-Hard*. We have also prepared a video that records the replays in these two environments. Please refer to the supplemental materials.

Regarding the *3s5z_vs_3s6z* challenge, the allied army owns three Stalkers and five Zealots, and the enemies have one more Zealot by comparison. From what Figure 16 has shown, the major issue that the agents need to learn and solve is how to break through the enhanced protection for the Stalkers made by the six opposing Zealots. Through the battle, our Stalkers attempt to lure the enemy Zealots, and our Zealots focus fire on the opposing Stalkers at the same time. In the end, the two live allied Zealots destroy the last opponent. Every MARL baseline performs poorly in this challenge. For example, VDN repeatedly fails to control the units to protect the allied Stalkers properly and is crushed by the opposing Zealots, as shown by Figure 17.

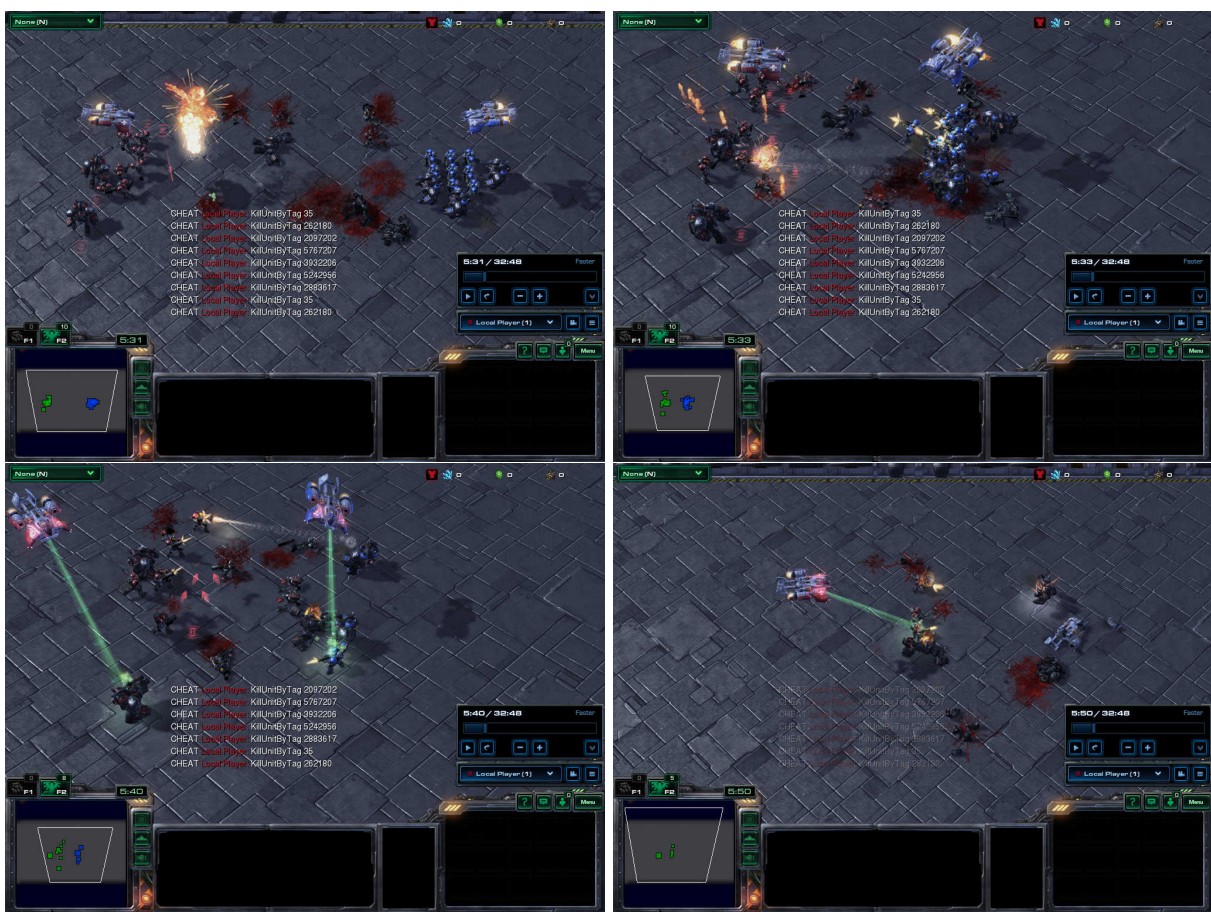

Figure 18: Our model VDFD in *MMM2*.

Furthermore, we look into the scenario of *MMM2*. The key tactic for the VDFD model to learn in this challenge is to annihilate the Medivac in the enemy army as soon as possible because it can heal the three Marauders and the eight Marines. Apparently, we have the privilege after the opposing Medivac is obliterated, and our army is still left with the Medivac, three Marines, and one Marauder when the enemies get defeated. In contrast, Figure 19 discloses the battle scenes of VDN. Because the agents failed to prioritize the elimination of the opposing Medivac, it healed the enemy Marauders to sustain the hits from the allies and ruined them. All the opposing Marauders survived at last.

In conclusion, we have illustrated that VDFD achieves state-of-the-art performance using the combat scenes captured within the game, surpassing the baselines that lack model disentanglement. By providing an analy-

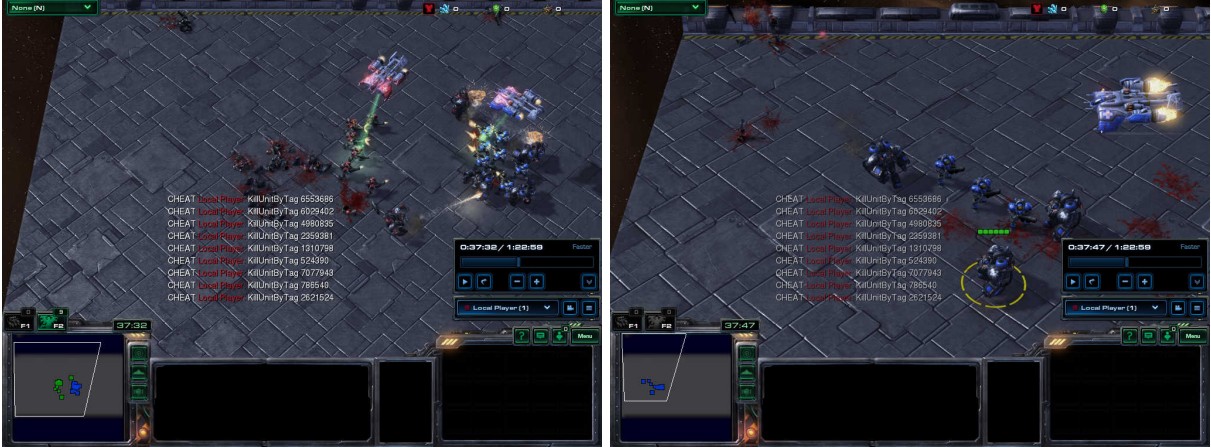

Figure 19: The VDN baseline in *MMM2*.

sis from the perspective of StarCraft II gameplay, we further strengthen our statement on the competitiveness of our approach.

## B.3    Performance of All Tested Algorithms in Multi-Agent MuJoCo Environments

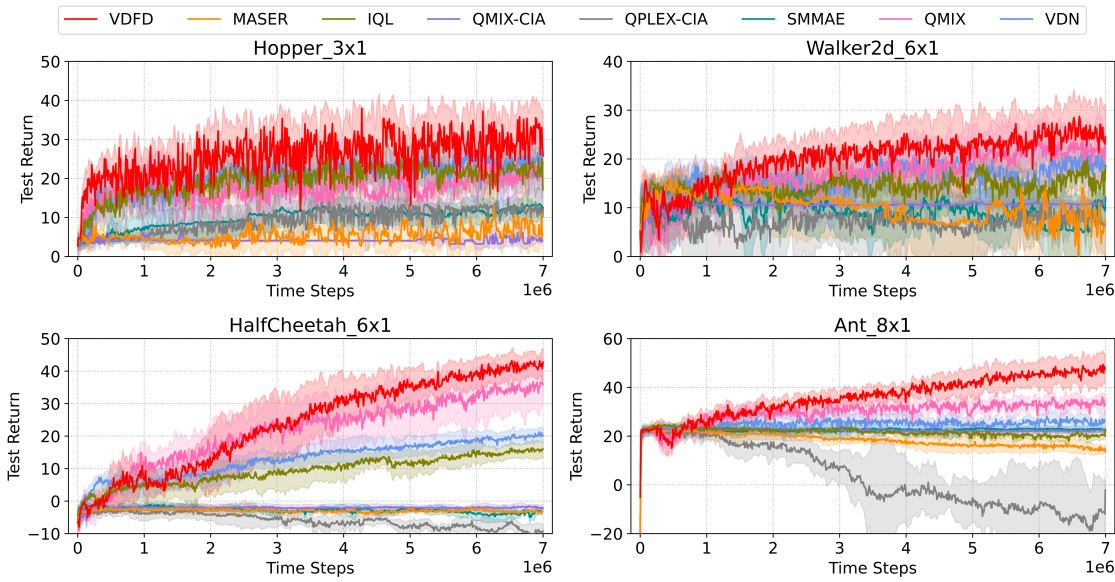

Figure 20: Episodic return of VDFD against the other MARL algorithms including QMIX-CIA, QPLEX-CIA, MASER, QMIX, VDN, IQL, and SMMAE in Multi-Agent MuJoCo environments. For clear plotting, the values of the returns are scaled for all methods, and the details can be found in Appendix C. VDFD outperforms all other algorithms in Multi-Agent MuJoCo.

## B.4 Performance of VDFD Compared to Baselines in Level-Based Foraging Tasks

Figure 21: Episode return of VDFD compared with the other MARL baselines including QMIX, VDN, IQL, SMMAE, and COMA in Level-Based Foraging environments. We select four representative tasks in which the levels of partial observability and cooperation vary, and in which the numbers of players and food items differ. The outstanding competence of VDFD is evident in all of the learning tasks. In addition, it is interesting to observe that QMIX and VDN can beat SMMAE, especially in environments that require full cooperation.

## B.5 Performance Comparison Between VDFD and Another Model-Based MARL Algorithm

Table 6: To complement the results in Section 5 of the main paper, we decide to compare the performance of VDFD with MAG (Models as AGents), a model-based MARL algorithm that was recently proposed (Wu et al., 2023). This table displays the mean and standard deviation of the Test Win Rate (%) in SMAC (StarCraft II Multi-Agent Challenge) and the episodic return in Multi-Agent MuJoCo (without scaling) for VDFD and MAG. During the experiments, the total number of roll-outs in a single run is the same for both VDFD and MAG. We make sure that a fair comparison is made in this way. From the results, we can observe that VDFD achieves better performance in seven of the eight environments overall. VDFD is very consistent because it always yields lower standard deviations than MAG across the runs.

| Environments \Methods | VDFD (Ours) | MAG |
|---|---|---|
| *2c_vs_64zg* (SMAC) | $91.88 \pm 3.92$ | $31.67 \pm 18.62$ |
| *2s_vs_1sc* (SMAC) | $99.37 \pm 1.39$ | $95.00 \pm 4.08$ |
| *2s3z* (SMAC) | $98.75 \pm 1.71$ | $93.33 \pm 2.89$ |
| *3s_vs_3z* (SMAC) | $99.88 \pm 0.17$ | $97.50 \pm 2.89$ |
| *3s_vs_4z* (SMAC) | $99.08 \pm 0.92$ | $85.00 \pm 7.64$ |
| *3s_vs_5z* (SMAC) | $92.50 \pm 4.74$ | $51.25 \pm 23.11$ |
| *3s5z_vs_3s6z* (SMAC) | $17.19 \pm 2.20$ | $32.73 \pm 22.18$ |
| Humanoid (MAMuJoCo) | $549 \pm 33$ | $423 \pm 104$ |

## B.6 Performance Comparison Between VDFD and HPN-QMIX

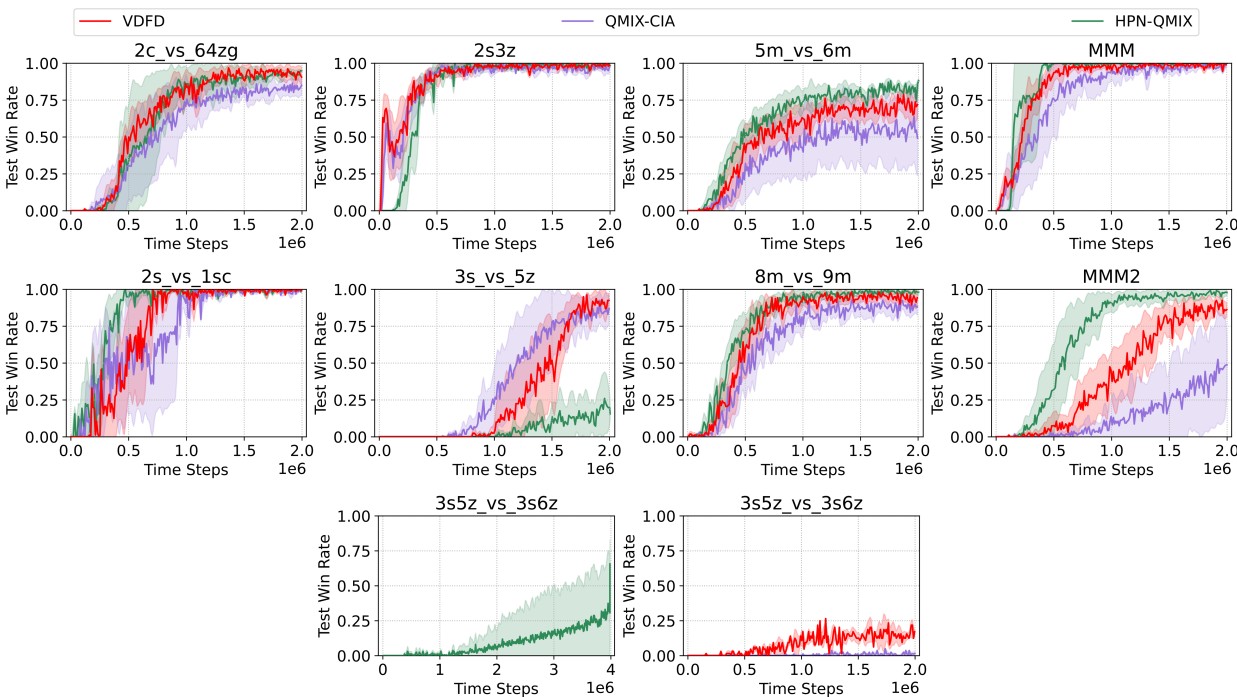

Figure 22: Test win rates of VDFD compared with QMIX-CIA (Liu et al., 2023) and HPN-QMIX (Jianye et al., 2023) in SMAC challenges. Both QMIX-CIA and HPN-QMIX boost the performance of QMIX by applying a newly proposed module to the QMIX framework. Each plot displays the mean and standard deviation over 5 seeds. It is important to note that the number of time steps is set to $2.0 \times 10^6$ in every environment except *3s5z__vs__3s6z*. In *3s5z__vs__3s6z* it is set to $4.0 \times 10^6$ for HPN-QMIX. This is different from the original paper (Jianye et al., 2023), in which the number of time steps is set to $1.0 \times 10^7$ for HPN-QMIX. The reason is that the number of time steps for the SMAC experiments in our paper is always set to $2.0 \times 10^6$ and we would like to maintain the consistency in the experiments with HPN-QMIX. VDFD and HPN-QMIX yield the same level of performance in most of the challenges. In both *3s__vs__5z* and *3s5z__vs__3s6z*, the win rates of HPN-QMIX increment slowly during the first $2.0 \times 10^6$ time steps, exceeded by our VDFD method. After running for $3.5 \times 10^6$ time steps in *3s5z__vs__3s6z*, HPN-QMIX rises above 25% and spikes after some more steps. These observations are consistent with the plots shown in (Jianye et al., 2023).

## C  Experimental Details

| Name | Value |
|------|-------|
| Action selector | Epsilon-greedy |
| Number of time steps | 2000000 |
| Agent architecture | RNN |
| Batch size | 32 |
| Buffer size | 5000 |
| Number of environments to run in parallel | 1 |
| Branching factor | 3 |
| Learning rate | $5 \times 10^{-4}$ |
| Discount hyperparameter $\gamma$ | 0.99 |
| Optimizer | RMSProp |
| RMSProp hyperparameter $\alpha$ | 0.99 |
| RMSProp hyperparameter $\epsilon$ | $1 \times 10^{-5}$ |
| Dimension of the hidden state for agent network | 64 |
| Latent dimension of an agent | 16 |
| Dimension of the embedding layer for agent network | 4 |
| Dimension of the embedding layer for mixing network | 32 |
| Dimension of the embedding layer for hyper-network | 64 |
| Number of hyper-network layers | 2 |
| Runner | Episodic runner |
| Logging interval for runner | $1 \times 10^4$ |
| Logging interval for training | $1 \times 10^4$ |
| Logging interval for the summary of statistics | $1 \times 10^4$ |
| Starting value of exploration rate | 1 |
| Finishing value of exploration rate | 0.05 |
| Number of time steps for epsilon rate annealing | $5 \times 10^4$ |
| Number of time steps after which the target networks are updated | 200 |
| Roll-out horizon | 3 |
| Loss function parameters $\beta_1, \beta_2, \beta_3, \beta_4$ | 1/2 |
| Loss function parameter $\beta_5$ | 1/3 |
| Whether to use greedy evaluation | True |
| Whether to use KL balancing | True |
| KL balancing $\alpha$ | 0.8 |
| Number of time steps between test intervals | 10000 |
| Number of episodes for testing every time | 32 |

Table 7: The set-up of hyperparameters for our VDFD method. The hyperparameters are used for all the benchmark environments, with the exceptions that the number of time steps for MAMuJoCo was set to 7M and the number of time steps for the ablation experiments in the *MMM2* environment (Section 5.4) was set to 3M so that the performance of the ablations can plateau.

In Table 7 and Table 8, we show the hyperparameter settings for the VDFD model and the environmental set-up for SMAC, respectively. The settings for Multi-Agent MuJoCo and for Level-Based Foraging are displayed in Table 9 and Table 10. To make clear graphs for Multi-Agent MuJoCo, we scale the returns for the four tasks in MAMuJoCo. We have specified this in Table 9.

Empirically, we had some hyperparameter tuning to choose the values of the real-valued masks $\mathbf{M}^{\mathrm{ac}}, \mathbf{M}^{\mathrm{af}}, \mathbf{M}^{\mathrm{st}}$ within the range of $[0, 1]$. In Section 5, we have specified that $\beta_1$ to $\beta_5$ are hyperparameters corresponding to the real-valued masks. In practice, we choose the value in $\mathbf{M}^{\mathrm{ac}}$ to be 1/2, $\mathbf{M}^{\mathrm{af}}$ to be 1/2, and $\mathbf{M}^{\mathrm{st}}$ to be 1/3, consistent with the values of the $\beta$'s in Table 7.

The hyperparameters of the other MARL baselines are consistent with their official implementations. Our implementation for VDFD is based on the Pytorch framework. The version of Python is 3.9.16. We run each

of the experiments in this paper with 5 different seeds. We use Linux machines with the Intel(R) Xeon(R) W-2133 CPU and the NVIDIA GeForce RTX 3090 GPU.

| Description | Value |
|---|---|
| Whether to consider episodes continuing or finished after the time limit is reached | False |
| Whether agents receive the last actions of all units within the sight range as part of observations | False |
| Whether agents receive pathing values within the sight range as part of observations | False |
| Whether to log messages about observations, states, actions, and rewards | False |
| Whether to use a combination of agents' observations as the global state | False |
| Whether agents receive the health of other agents within the sight range | True |
| Whether to use a non-learning heuristic AI | False |
| Whether agents receive their own health | True |
| Whether to scale down the reward for every episode | True |
| Number of time steps | 2000000 |
| Reward scaling rate | 20 |
| Reward for winning (all enemies die) | +200 |
| Reward for defeating one enemy | +10 |
| Reward when one ally dies | −5 |
| Number of time steps per agent step | 8 |
| How far away units move per step | 2 |
| Races | {Random, Protoss, Terran, Zerg} |
| Basic Actions | {attack, move, stop, heal, no-op} |
| Directions | {East, West, South, North} |
| Difficulty | 7 (very difficult) |
| Game version | SC2.4.10 |

Table 8: The default settings for SMAC environments.

| Name | Value |
|---|---|
| Number of time steps | 7000000 |
| Number of nearest joints to observe | 2 |
| Batch size | 32 |
| Number of episodes for testing during the test interval | 32 |
| Number of time steps between testing intervals | 10000 |
| Number of time steps between logging intervals | 10000 |
| Whether the agent receives its ID | True |
| Whether agents receive the last actions within the sight range | True |
| Partitioning of the robot *Ant* | (hip4, ankle4, hip1, ankle1, hip2, ankle2, hip3, ankle3) |
| Partitioning of the robot *Half Cheetah* | (bthigh, bshin, bfoot, fthigh, fshin, ffoot) |
| Partitioning of the robot *Hopper* | (thigh_joint, leg_joint, foot_joint) |
| Partitioning of the robot *Walker2d* | (foot_joint, leg_joint, thigh_joint, foot_left_joint, leg_left_joint, thigh_left_joint) |
| Scaling factor for the returns of *Ant_8×1* | 0.2 |
| Scaling factor for the returns of *HalfCheetah_6×1* | 0.2 |
| Scaling factor for the returns of *Hopper_3×1* | 0.1 |
| Scaling factor for the returns of *Walker2d_6×1* | 0.1 |

Table 9: The default settings for Multi-Agent MuJoCo control suite.

| Name | Value |
|---|---|
| Number of time steps | 2000000 |
| Number of episodes for testing during the test interval | 100 |
| Number of time steps between testing intervals | 50000 |
| Number of time steps between logging intervals | 50000 |
| Basic Actions | {None, North, South, East, West, Load} |
| Whether the agent receives its ID | True |
| Whether agents receive the last actions within the sight range | True |

Table 10: The default settings for Level-Based Foraging environments.

# D    Markov Games

In MARL environments, the dynamics of the environment and the reward are determined by the joint actions of all agents, and a generalization of MDP that is able to model the decision-making processes of multiple agents is needed. This generalization is known as stochastic games, also referred to as Markov games (Littman, 1994).

**Definition D.1.** A Markov game $G_M$ is defined as a tuple

$$\langle \mathcal{N}, \mathcal{S}, \{\mathcal{A}^i\}_{i \in \{1,...,N\}}, P, \{\mathbf{R}^i\}_{i \in \{1,...,N\}}, \gamma \rangle, where$$

- $\mathcal{N} = \{1, ..., N\}$ represents the set of $N$ agents, and it is equivalent to a single-agent MDP when $N = 1$.

- $\mathcal{S}$ is the state space of all agents in the environment.

- $\{\mathcal{A}^i\}_{i \in \{1,...,N\}}$ stands for the set of action space of every agent $i \in \{1, ..., N\}$.

- Define $\mathcal{A} := \mathcal{A}^1 \times ... \times \mathcal{A}^N$ to be the set of all possible joint actions of $N$ agents, and define $\triangle(\mathcal{S})$ to be the probability simplex on $\mathcal{S}$. $P : \mathcal{S} \times \mathcal{A} \to \triangle(\mathcal{S})$ is the probability function that outputs the transition probability to any state $s' \in \mathcal{S}$ given state $s \in \mathcal{S}$ and joint actions $\mathbf{a} \in \mathcal{A}$.

- $\{\mathbf{R}^i\}_{i \in \{1,...,N\}}$ is the set of reward functions of all agents, in which $\mathbf{R}^i : \mathcal{S} \times \mathcal{A} \times \mathcal{S} \to \mathbb{R}$ denotes the reward function of the $i$-th agent that outputs a reward value on a transition from state $s \in \mathcal{S}$ to state $s' \in \mathcal{S}$ given the joint actions of all agents $\mathbf{a} \in \mathcal{A}$.

- Lastly, $\gamma \in [0, 1]$ represents the discount factor w.r.t. time.

A Markov game, when used to model the learning of multiple agents, makes the interactions between agents explicit. At an arbitrary time step $t$ in the Markov game, an agent $i$ takes its action $a_{i,t}$ at the same time as any other agent given the current state $s_t$. The agents' joint actions, $\mathbf{a}_t$, cause the transition to the next state $s_{t+1} \sim P(\cdot|s_t, \mathbf{a}_t)$ and make the environment to generate a reward $\mathbf{R}^i$ for agent $i$. Every agent has the goal of maximizing its own long-term reward, which can be achieved by finding a behavioral policy

$$\pi(\mathbf{a}|s) := \prod_{i \in \mathcal{N}} \pi^i(a^i|s)$$

Specifically, the value function of agent $i$ is defined as

$$V^i_{\pi^i, \pi^{-i}}(s) := \mathbb{E}_{s_{t+1} \sim P(\cdot|s_t, \mathbf{a}_t), a^{-i} \sim \pi^{-i}(\cdot|s_t)} \Big[ \sum_t \gamma^t \mathbf{R}^i_t(s_t, \mathbf{a}_t, s_{t+1}) | a^i_t \sim \pi^i(\cdot|s_t), s_0 \Big]$$

where the symbol $-i$ stands for the set of all indices in $\{1, ..., N\}$ excluding $i$. It is clear that in a Markov game, the optimal policy of an arbitrary agent $i$ is always affected by not only its own behaviors but also the policies of the other agents. This situation gives rise to significant disparities in the approach to finding solutions between traditional single-agent settings and multi-agent reinforcement learning.

In the realm of MARL, a prevalent situation arises where agents do not have access to the global environmental state. They are only able to make observations of the state by leveraging an observation function. This scenario is formally defined as Dec-POMDP, as we have shown in **Definition 3.1**. It contains two extra terms for the observation function

$$O(s, i) : \mathcal{S} \times \mathcal{N} \to \mathcal{Z}$$

and for the set of observations $\mathcal{Z}$ made by each of the agents, in addition to the definition of the Markov game.

# E  Detailed Deduction of the ELBO

Equation 2 in the main paper briefly shows how the ELBO of Dec-POMDP, $\mathcal{L}_{\text{ELBO}}(\mathbf{a}_{0:T}, \mathbf{z}_{1:T})$, is derived. We present the full deduction process here.

$$
\begin{aligned}
&\mathcal{L}_{\text{ELBO}}(\mathbf{a}_{0:T}, \mathbf{z}_{1:T}) \\
&= \log p\left(\mathbf{a}_{0:T}, \mathbf{z}_{1:T}\right) \\
&= \log \mathbb{E}_{q_\theta(\hat{s}_{1:T}|\mathbf{a}_{0:T}, \mathbf{z}_{1:T})} \left[ \frac{p(\hat{s}_{1:T}, \mathbf{a}_{0:T}, \mathbf{z}_{1:T})}{q_\theta(\hat{s}_{1:T} \mid \mathbf{a}_{0:T}, \mathbf{z}_{1:T})} \right] \\
&\geq \mathbb{E}_{q_\theta(\hat{s}_{1:T}|\mathbf{a}_{0:T}, \mathbf{z}_{1:T})} \log \left[ \frac{p(\hat{s}_{1:T}, \mathbf{a}_{0:T}, \mathbf{z}_{1:T})}{q_\theta(\hat{s}_{1:T} \mid \mathbf{a}_{0:T}, \mathbf{z}_{1:T})} \right] \\
&= \int q_\theta(\hat{s}_{1:T} \mid \mathbf{a}_{0:T}, \mathbf{z}_{1:T}) \log \left[ \frac{p(\hat{s}_{1:T}, \mathbf{a}_{0:T}, \mathbf{z}_{1:T})}{q_\theta(\hat{s}_{1:T} \mid \mathbf{a}_{0:T}, \mathbf{z}_{1:T})} \right] d\hat{s}_{1:T} \\
&= \int \sum_{t=1}^{T} q_\theta(\hat{s}_{1:T} \mid \mathbf{a}_{0:T}, \mathbf{z}_{1:T}) \log \left[ \frac{p\left(\mathbf{a}_t \mid \mathbf{z}_t\right) p\left(\mathbf{z}_t \mid \hat{s}_t\right) p\left(\hat{s}_t \mid \hat{s}_{t-1}, \mathbf{a}_{t-1}\right)}{q_\theta\left(\hat{s}_t \mid \hat{s}_{t-1}, \mathbf{a}_{t-1}, \mathbf{z}_t\right)} \right] d\hat{s}_{1:T} \\
&= \sum_{t=1}^{T} \int q_\theta(\hat{s}_{1:t} \mid \mathbf{a}_{0:t}, \mathbf{z}_{1:t}) \log \left[ \frac{p\left(\mathbf{a}_t \mid \mathbf{z}_t\right) p\left(\mathbf{z}_t \mid \hat{s}_t\right) p\left(\hat{s}_t \mid \hat{s}_{t-1}, \mathbf{a}_{t-1}\right)}{q_\theta\left(\hat{s}_t \mid \hat{s}_{t-1}, \mathbf{a}_{t-1}, \mathbf{z}_t\right)} \right] d\hat{s}_{1:t} \\
&= \sum_{t=1}^{T} \Bigg\{ \int q_\theta(\hat{s}_{1:t} \mid \mathbf{a}_{0:t}, \mathbf{z}_{1:t}) \log \left[ p\left(\mathbf{a}_t \mid \mathbf{z}_t\right) p\left(\mathbf{z}_t \mid \hat{s}_t\right) \right] d\hat{s}_{1:t} \\
&\quad + \int q_\theta(\hat{s}_{1:t} \mid \mathbf{a}_{0:t}, \mathbf{z}_{1:t}) \log \left[ \frac{p\left(\hat{s}_t \mid \hat{s}_{t-1}, \mathbf{a}_{t-1}\right)}{q_\theta\left(\hat{s}_t \mid \hat{s}_{t-1}, \mathbf{a}_{t-1}, \mathbf{z}_t\right)} \right] d\hat{s}_{1:t} \Bigg\} \\
&= \sum_{t=1}^{T} \Bigg\{ \int q_\theta(\hat{s}_{1:t} \mid \mathbf{a}_{0:t}, \mathbf{z}_{1:t}) \log \left[ p\left(\mathbf{a}_t \mid \mathbf{z}_t\right) p\left(\mathbf{z}_t \mid \hat{s}_t\right) \right] d\hat{s}_{1:t} \\
&\quad - \int q_\theta\left(\hat{s}_{1:t-1} \mid \mathbf{a}_{0:t-1}, \mathbf{z}_{1:t-1}\right) \mathcal{D}_{\text{KL}}\left[ q_\theta\left(\hat{s}_t \mid \hat{s}_{t-1}, \mathbf{a}_{t-1}, \mathbf{z}_t\right) \,\|\, p\left(\hat{s}_t \mid \hat{s}_{t-1}, \mathbf{a}_{t-1}\right) \right] d\hat{s}_{1:t} \Bigg\} \\
&= \mathbb{E}_{q_\theta(\hat{s}_{1:T}|\mathbf{a}_{0:T}, \mathbf{z}_{1:T})} \sum_{t=1}^{T} \Big\{ \log \left[ p\left(\mathbf{a}_t \mid \mathbf{z}_t\right) p\left(\mathbf{z}_t \mid \hat{s}_t\right) \right] \\
&\quad - \mathcal{D}_{\text{KL}}\left[ q_\theta\left(\hat{s}_t \mid \hat{s}_{t-1}, \mathbf{a}_{t-1}, \mathbf{z}_t\right) \,\|\, p\left(\hat{s}_t \mid \hat{s}_{t-1}, \mathbf{a}_{t-1}\right) \right] \Big\} \\
&\approx \sum_{t=1}^{T} \Big\{ \log \left[ p\left(\mathbf{a}_t \mid \mathbf{z}_t\right) p\left(\mathbf{z}_t \mid \hat{s}_t\right) \right] \\
&\quad - \mathcal{D}_{\text{KL}}\left[ q_\theta\left(\hat{s}_t \mid \hat{s}_{t-1}, \mathbf{a}_{t-1}, \mathbf{z}_t\right) \,\|\, p\left(\hat{s}_t \mid \hat{s}_{t-1}, \mathbf{a}_{t-1}\right) \right] \Big\} \\
&= \sum_{t=1}^{T} \Big\{ \log \left[ p\left(\mathbf{a}_t \mid \mathbf{z}_t\right) \right] + \log \left[ p\left(\mathbf{z}_t \mid \hat{s}_t\right) \right] \\
&\quad - \mathcal{D}_{\text{KL}}\left[ q_\theta\left(\hat{s}_t \mid \hat{s}_{t-1}, \mathbf{a}_{t-1}, \mathbf{z}_t\right) \,\|\, p\left(\hat{s}_t \mid \hat{s}_{t-1}, \mathbf{a}_{t-1}\right) \right] \Big\}
\end{aligned}
$$

(5)

(6)

(7)

Note that (5) is reached via Jensen's inequality. Equation 6 can be obtained by sampling $\hat{s}_{1:T} \sim q_\theta\left(\hat{s}_{1:T} \mid \mathbf{a}_{0:T}, \mathbf{z}_{1:T}\right)$. Equation 7 is identical to Equation 2.

# F  Supplementary Related Work

## F.1  Variational Auto-Encoder (VAE)

The VAE model is a generative model that learns a probability distribution across the input space (Fortuin et al., 2019). It is composed of an encoder network and a decoder network. When provided with the input data $\mathbf{x}$, VAE assumes that $\mathbf{x}$ is generated from a latent variable $\mathbf{z}$ that is not directly observed (Kingma & Welling, 2013). The latent variable $\mathbf{z}$ is sampled from the prior distribution over the latent space, which is the centered isotropic multivariate Gaussian $p(\mathbf{z}) = \mathcal{N}(\mathbf{z}; \mathbf{0}, \mathbf{I})$. The VAE then proceeds to learn the conditional distribution $p(\mathbf{x}|\mathbf{z})$. The posterior distribution of the latent variables, $p(\mathbf{z}|\mathbf{x})$, is assumed to take on an approximate Gaussian with a diagonal covariance to address the intractability of the true posterior distribution. Consequently, the reconstruction error can be computed and back-propagated through the encoder-decoder network (Ha & Schmidhuber, 2018).

The encoder of the VAE can be considered as a recognition model, and the decoder serves as a generative model (Kingma & Welling, 2013). The VAE model can then be described as the combination of two coupled but independently parameterized models. These two models support each other: the recognition model provides the generative model with the approximates for its posteriors over latent random variables. And conversely, the generative model enables the recognition model to learn informative representations of the data. The recognition model is the approximate inverse of the generative model, according to Bayes' rule.

## F.2  Policy Gradient MARL Methods

In single-agent RL, there exists a class of policy gradient methods that do not necessarily require estimates of the value functions. These algorithms update the learning parameters along the direction of the gradient of specific metrics with respect to the policy parameter (Sutton & Barto, 2018). The optimal policy is estimated using parametrized function approximations.

Policy gradient algorithms belong to one of the two main categories of MARL algorithms, including actor-critic methods that update policy networks while learning a centralized value function to guide policy optimization based on the policy gradient theorem (Yang & Wang, 2020). Gupta et al. (2017) described a multi-agent policy gradient version of the trust region policy optimization (Schulman et al., 2015) that enables policy parameter sharing among all agents, but the actor and the critic can only be conditioned on local observations and actions. BiCNet (Peng et al., 2017) also allowed parameter sharing to enhance the scalability of the model, but the communication among agents actually depends on bi-directional RNN. Lowe et al. (2017) adopted the framework of deep deterministic policy gradient (Lillicrap et al., 2015) to multi-agent settings and proposed MADDPG, an approach that uses actors trained on the local observations and a centralized critic for function approximation. The critic is learned by the agents based on their joint observation and joint action. COMA (Foerster et al., 2018) utilized a critic for centralized learning as well, conditioning the critic on the agents' actions and global state information. The most notable feature of COMA is that the critic computes a counterfactual baseline, to which the estimated return for the joint action is compared. However, COMA tends to suffer high variance in the computation of the counterfactual baseline, causing instabilities in multi-agent benchmarking (Papoudakis et al., 2020). Yu et al. (2021) carefully investigated the performance of proximal policy optimization (Schulman et al., 2017) in cooperative multi-agent environments and obtained competitive sample efficiency with minimal hyperparameter tuning and no major algorithmic modifications. Designing a centrally computed critic to pass the current state information into decentralized agents for learning optimal cooperative behaviors has proved to be an important line of approach in addressing the credit assignment problem (Iqbal & Sha, 2018; Du et al., 2019; Zhou et al., 2020).

# G    Limitations and Broader Impact

We acknowledge the numerical nature of our paper. Our research primarily emphasizes the practical application and evaluation of the proposed disentangled representation learning approach. Our focus on experiments in various benchmarks is intentional, as we aim to empirically demonstrate the generalizability and robustness of our method. While we recognize the significance of theoretical guarantees in MARL, the complexities of multi-agent systems and our method present great challenges in deriving rigorous mathematical proofs. Addressing the theoretical side of disentangled representation learning could be a focus of our future work.

Since many hyperparameters are introduced in our work, hyperparameter tuning may be needed when VDFD is applied to new multi-agent environments. The computational cost of the method can increase in this case. The performance may vary based on the choices of hyperparameters. We did not obtain very poor results, and the reason could be that we did not set the hyperparameters to extreme values.

Our work aims at contributing to the development of multi-agent and model-based RL algorithms. Although this development could be applied in various fields, it does not require specific ethical considerations to be highlighted. Nonetheless, it is crucial to emphasize the importance of responsible deployment to ensure a beneficial impact on society.

