# OpenReview forum: "Leveraging World Model Disentanglement in Value-Based Multi-Agent Reinforcement Learning"
_TMLR — Rejected by TMLR_

### Review · Reviewer_mLpZ · 2024-06-17

**Summary Of Contributions:**

This paper proposed a model-based multi-agent reinforcement learning (MARL) approach that integrates a world model providing rollout information to augment the QMIX method. The proposed world model highlights a disentanglement involving action-conditioned, action-free, and static branches, which aids in modeling the partial-observable, non-stationary environment typical in MARL settings. Experiments conducted on SMAC, MAMuJoCo, and LBF demonstrate the superiority of the proposed VDFD method over various baselines and validate the effectiveness of its main components.

**Audience:**

Yes

**Claims And Evidence:**

No

**Requested Changes:**

Please refer to the Weaknesses section for detailed suggestions, with particular emphasis on improving writing and elaborating on algorithmic details. The authors should redraw Figures 1 and 2 for a clearer presentation.

**Strengths And Weaknesses:**

Strengths:

- Introducing MBRL into MARL is a promising direction.
- The idea of disentanglement is important for handling partial-observable, non-stationary environments in MARL
- Extensive experiments are conducted to demonstrate the effectiveness of the proposed method.

Weaknesses:

- Clarity: The paper lacks detailed and clear descriptions of its approach. Many improvements are needed, including but not limited to:
  - Architecture:
    - The input of the Graph VAE (GALA in this paper) is not specified, and the exact formulation of $h_t^i$ and $\mathbf{h}_t$ in Section 4.1.3 is unclear in both the main text and the appendix.
    - In Section 4.1.4, $\mathbf{M}^{\mathrm{ac}}, \mathbf{M}^{\mathrm{af}}, \mathbf{M}^{\text{st}}$ are used to aggregate the three outputs of the branches and are described as hyper-parameters. Are these masks predefined without training? However, in Section 4.2, they are used to aggregate sequences of states $\left\{\hat{s}_t, \ldots, \hat{s}_{t+j}\right\},\left\{\hat{v}_t, \ldots, \hat{v}_{t+j}\right\}$ along the temporal dimension. It is unclear how the same masks can be used for these distinct purposes.
    - In Figure 1, the architecture of VAE is not clearly illustrated. What are the input and output of the orange box on the left? Which parameters are shared among the three modules on the right?
    - In Figure 2, why $\hat{s}\_t$ is directly linked to $\hat{s}\_{t+1}, \hat{s}\_{t+2}, \hat{s}\_{t+3}$?
  - The training of VDFD:
    - Are all models, including the world model and agent network, jointly updated according to Equation (4), or is the training of the model and agent interleaved?
    - In Figure 2, are gradients backpropagated along the rollout to accumulate on the agent network?
  - The beginning of Section 4.1 introduces two kinds of latent states, $s_t$ and $v_t$, but Section 4.1.1 only mentions $s_t$. It seems that Section 4.1.1 derives the objective without disentanglement, which is later introduced into the learning objective in Section 4.3. This is confusing on the first read.
  - Minor:
    - Typo: In Section 3.1, $\tau^i \in \mathcal{T}:=(\mathcal{Z} \times \mathcal{A})^*$  and $\pi^i\left(a^i \mid \tau^i\right): \mathcal{T} \times \mathcal{A} \rightarrow[0,1]$ should be $\tau^i \in \mathcal{T}^i:=(\mathcal{Z} \times \mathcal{A}^i)^*$ and $\pi^i\left(a^i \mid \tau^i\right): \mathcal{T}^i \times \mathcal{A}^i \rightarrow[0,1]$.
    - Notation: The use of $q_\pi$ for the optimal joint policy can be confused with the posterior $q$.
- Related work: The paper lacks a sufficient literature review on model-based reinforcement learning. The proposed method, which uses rollout information to augment a model-free method like QMIX, is more similar to [1] than to the cited works PlaNet and Dreamer, which train policies entirely using on-policy imaginary rollouts (i.e., a Dyna-style algorithm).
- Insufficient analysis of disentanglement: This paper only provides numerical results to demonstrate the effectiveness of the modules in disentanglement. Additional analysis, such as visualization or case studies, is needed to show that the model indeed learns disentanglement.

[1] Racanière, Sébastien, et al. "Imagination-augmented agents for deep reinforcement learning." Advances in neural information processing systems 30 (2017).

---

> ### Author Response · Authors · 2024-07-09
> **Authors' Response to Reviewer mLpZ (1)**
>
> Dear reviewer,
>
> We sincerely appreciate your comments and suggestions. Please take a look at our response to your questions below. Following the TMLR guidelines, we will provide a revised version of the manuscript after all reviewers have submitted their reviews.
>
> - **The input of the Graph VAE (GALA ) is not specified, and the exact formulation of $h_t^i$ and $\mathbf{h}_t$ in Section 4.1.3 is unclear.**
>
> A: Thanks for pointing out. In addition to our response here, we will modify Figures 1(a) and 1(b) to specify the input of GALA. We will also provide the formulation of $h_t^i$ and $\mathbf{h}\_t$ in the updated manuscript.
>
> In the action-conditioned branch, the input to GALA is $\mathbf{h}\_t$. In the action-free branch, the input to GALA is $(\hat{v}\_{t-1}, \mathbf{z}\_{t})$; in the static branch it is $\mathbf{z}\_{1:t}$.
>
> In Section 4.1.3, we mention that using the re-parametrization technique, we can formulate $q\_{\theta}(\cdot | \hat{s}\_{t-1}, \mathbf{a}\_{t-1}, \mathbf{z}\_{t} )$ as $q\_{\theta}(\cdot | \mathbf{h}\_t )$. Specifically, we use a neural network $f\_{\theta}(\hat{s}\_{t-1}, \mathbf{a}\_{t-1}, \mathbf{z}\_{t} )$ to extract low dimensional features $\omega\_t$, i.e., $\omega\_t = f\_{\theta}(\hat{s}\_{t-1}, \mathbf{a}\_{t-1}, \mathbf{z}\_{t} )$. Then we use a recurrent neural network to encapsulate the past information of the agents, i.e., $\mathbf{h}\_t = rnn(\omega\_t, \mathbf{h}\_{t-1})$ and $\mathbf{h}\_0 = \hat{s}\_{0}$. This formulation also holds for each $h\_t^i$.
>
> - **In Section 4.1.4, $\mathbf{M}^{\mathrm{ac}}, \mathbf{M}^{\mathrm{af}}, \mathbf{M}^{\text{st}}$ are used to aggregate the three outputs of the branches and are described as hyper-parameters. Are these masks predefined without training? In Section 4.2 they are used to aggregate sequences of states along the temporal dimension. It is unclear how the same masks can be used for these distinct purposes.**
>
> A: Thanks for asking. In Section 4.1.4, $\mathbf{M}^{\mathrm{ac}}, \mathbf{M}^{\mathrm{af}}, \mathbf{M}^{\text{st}}$ are real-valued masks that are pre-defined without training. In Section 4.2, what we describe is basically the same as in 4.1.4 except that we now perform a certain number of roll-outs (therefore the roll-out horizon is set to $j$). The sequences of states $(\hat{s}\_t, \ldots, \hat{s}\_{t+j})$ and $(\hat{v}\_t, \ldots, \hat{v}\_{t+j})$ are the outputs of the action-conditioned and the action-free branches, respectively. They can also be expressed as $\mathbf{o}^{\mathrm{ac}}$ and $\mathbf{o}^{\mathrm{af}}$. The output of the static branch is $\hat{\mathbf{z}}^{\mathrm{static}}$.
>
> The transformation of the outputs of the three branches can be written as $\hat{s\_{t}}^{\mathrm{Rollout}} = \mathbf{M}^{\mathrm{ac}} \odot \mathbf{o}^{\mathrm{ac}}$,
> $\hat{v\_{t}}^{\mathrm{Rollout}} = \mathbf{M}^{\mathrm{af}} \odot \mathbf{o}^{\mathrm{af}}$,
> and
> $\hat{\mathbf{z}}^{\mathrm{Rollout}} = \mathbf{M}^{\mathrm{st}} \odot \hat{\mathbf{z}}^{\mathrm{static}}$. This shows that the transformations in 4.2 are consistent with 4.1.4 and that the masks are indeed used for the same purpose.
>
> We will also make clarifications in the updated manuscript regarding this question.
>
> - **In Figure 1, the architecture of VAE is not clearly illustrated. What are the input and output of the orange box on the left? Which parameters are shared among the three modules on the right?**
>
> A: Thanks for pointing out. We will modify Figures 1(a) and 1(b) to display the architecture more clearly in our updated manuscript.
>
> The orange box in the original Figure 1(a) denotes the standard VAE that is used for learning the prior distribution. The input to the VAE encoder is $(\hat{s}\_{t-1}, \mathbf{a}\_{t-1})$, and the output from the VAE decoder is $(\hat{s}\_{t-1}', \mathbf{a}\_{t-1}')$.
>
> Figure 1(b) is the right part of Figure 1. The parameters shared among the three modules of the world model are $\hat{s}\_{t-1}, \mathbf{a}\_{t-1}, \mathbf{h}\_t, \hat{v}\_{t-1}, \mathbf{z}\_{1:t} $.
>
> - **In Figure 2, why $\hat{s}\_t$ is directly linked to $\hat{s}\_{t+1}, \hat{s}\_{t+2}, \hat{s}\_{t+3}$?**
>
> A: Thanks for pointing out. $\hat{s}\_t$ is directly linked to $\hat{s}\_{t+1:t+3}$ because Figure 2 is a demonstration on the architecture of VDFD when the roll-out horizon is set to 3 (i.e., $j=3$). We link the latent states $\{\hat{s}\_t, \ldots, \hat{s}\_{t+3}\}$ together to show that they are aggregated and transformed into $\hat{s\_{t}}^{\mathrm{Rollout}}$ via the mask $\mathbf{M}^{\mathrm{ac}}$.
>
> We will also modify Figure 2 in our updated manuscript to clarify this question.

---

> ### Author Response · Authors · 2024-07-09
> **Authors' Response to Reviewer mLpZ (2)**
>
> **The training of VDFD:**
>
> - **Are all models, including the world model and agent network, jointly updated according to Equation (4), or is the training of the model and agent interleaved?**
>
> A: Thanks for asking. Realistically, the model and the agent network can be updated asynchronously. This is because the size of the agent network becomes very large in certain multi-agent settings, where the number of agents may exceed 50. The agent network will be updated after a certain number of episodes. The training of the model and the agents can be considered as interleaved.
>
> We will also clarify this question in our updated manuscript.
>
> - **In Figure 2, are gradients backpropagated along the rollout to accumulate on the agent network?**
>
> A: Thanks for asking. Because our model needs to operate in multi-agent settings with potentially large numbers of agents, we have disabled the calculation of gradients along the simulated roll-outs. We can reduce unnecessary memory consumption for computations like gradient back-propagation in the process of world model learning.
>
> We will also clarify this question in our updated manuscript.
>
> - **It seems that Section 4.1.1 derives the objective without disentanglement, which is later introduced into the learning objective in Section 4.3. This is confusing on the first read.**
>
> A: Thanks for pointing out. We will clarify this confusion in our updated manuscript.
>
> - **Typo in Section 3.1.**
>
> A: Thanks for the note. We will correct this typo in our updated manuscript.
>
> - **Notation: The use of $q\_\pi$ for the optimal joint policy can be confused with the posterior $q$.**
>
> A: Thanks for pointing out. We will add a note in our updated manuscript.
>
> - **Suggestions on Related Work.**
>
> A: Thanks for the feedback. We will elaborate on the literature review on model-based RL as suggested in our updated manuscript.
>
> - **Insufficient analysis of disentanglement.**
>
> A: Thanks for the advice. We will improve Appendix B in our updated manuscript to address this.
>
> - **Requested Changes: Please refer to the Weaknesses section with particular emphasis on improving writing and elaborating on algorithmic details. The authors should redraw Figures 1 and 2 for a clearer presentation.**
>
> A: Following the guidelines, we will provide a revised version of the manuscript after all reviewers have submitted their reviews. In the meantime, please let us know if you have any questions regarding our response.

---

> > ### Comment · Reviewer_mLpZ · 2024-08-29
> >
> > Thank you for the detailed response. After carefully reviewing the revised method section, I see that the clarity has improved significantly. However, there is still room for further enhancement. There are too many notations and notes to clarify these. The figures can be more informative, too. More importantly, the author should clearly separate their technical contributions from those adopted from the literature. For instance, content like ELBO, which appears similar to those in PlaNet and Dreamer, would be better placed in the Background section. Additionally, the idea of disentanglement seems highly relevant to Pan et al., but this connection is not sufficiently discussed.
> >
> > I cannot find the disentanglement analysis in Appendix B. Could the author clarify which section covers this and briefly describe what was done?

---

> > > ### Author Response · Authors · 2024-09-01
> > >
> > > Dear reviewer,
> > >
> > > Thank you for your thorough review and valuable feedback on our revised paper. We acknowledge your suggestions for further enhancement, particularly regarding the importance of clearly distinguishing our contributions from those adopted from previous literature. We will move the content of ELBO to the Background section and ensure that the connections to the work of Pan et al. are more explicitly discussed.
> > >
> > > Regarding the disentanglement analysis, we might not have fully understood your comment. In Appendix B.2, we present case studies in the StarCraft II benchmark to demonstrate how our model with disentanglement outperforms other methods without disentanglement. We provide visualization of the experiments, including gameplay images and a video in the supplemental materials, for the analysis. However, we are unsure if you expect additional types of analyses, such as visualizing the ablations of the disentangled model in Section 5.4. It would be nice if you could help us clarify this point.
> > >
> > > We plan to submit a newer version of the paper incorporating your suggestions after all three reviewers have read the revised paper and posted their comments. We believe this will help us address all feedback comprehensively and avoid any confusion for the other reviewers.

---

> > > > ### Comment · Reviewer_mLpZ · 2024-09-02
> > > >
> > > > My question is how you confirm that your model learns disentangled information (controllable, uncontrollable, and static) in your world model. An analysis like Fig. 3 in Pan et al. should be helpful.

---

### Review · Reviewer_FHth · 2024-07-17

**Summary Of Contributions:**

This paper studies multi-agent reinforcement learning based on value-function learning. It decomposes the value function in such a way that each agent has her own value function. Furthermore, it proposes an approach to learn a representation of the environment that is split into three parts: one part captures stationary features, one part corresponds to dynamic features that do not depend on the agent's actions, and the last part deals with the influence of the agent's actions on the environment. In this way, there is a form of disentanglement. Experiments on several complex environments are presented.

**Audience:**

Yes

**Broader Impact Concerns:**

Nothing specific.

**Claims And Evidence:**

Yes

**Requested Changes:**

1) Page 4: In the sentence before Section 3.2, could you please clarify what the reward $r_{t+k}$ can depend on?

2) Page 4, Definition 3.2: Under which assumptions on the model can we guarantee that the Individual-Global-Max (IGM) condition is satisfied?

3) Page 8: Why taking a Normal distribution for the reference distribution $p$?

4) Page 9, equation (4): The overall loss contains many terms and it might be difficult to know how to weigh the importance of each term. Do you have guidelines to chose the values of the coefficients $\beta_i$? Did you ever obtained very poor results for some “bad” choices of $\beta$'s?

5) Is the code going to be made available?

**Strengths And Weaknesses:**

Strengths: The ideas are explained quite clearly. As far as I know, the approach is new and seems to address an important question in MARL with an interesting method. The experiments are done on three complex models.

Weaknesses: The paper is purely numerical but I imagine it is hard to provide theoretical guarantees since the proposed approach is quite complex. The discussion on the limitations could be more detailed.

---

> ### Author Response · Authors · 2024-07-18
> **Authors' Response to Reviewer FHth (1)**
>
> Dear reviewer,
>
> We sincerely appreciate your insightful comments. Please take a look at our response to your questions below. Following the TMLR guidelines, we will provide a revised version of the manuscript after all reviewers have submitted their reviews.
>
> **Weaknesses: The paper is purely numerical but it is hard to provide theoretical guarantees since the proposed approach is quite complex. The discussion on the limitations could be more detailed.**
>
> A: Thanks for pointing out. We acknowledge the reviewer's observation regarding the numerical nature of our paper. Indeed, our research primarily emphasizes the practical application and evaluation of the proposed disentangled representation learning approach. Our focus on experiments in various benchmarks is intentional, as we aim to empirically demonstrate the generalizability and robustness of our method. While we recognize the importance of theoretical guarantees in MARL, the complexities of multi-agent systems and our method present significant challenges in deriving rigorous mathematical proofs.
>
> We concur that a more detailed discussion of the limitations could enhance the overall quality of the paper. We will add a section on limitations and broader impact in the appendix of the updated manuscript.
>
> **Requested Changes:**
>
> **1. Page 4: In the sentence before Section 3.2, could you please clarify what the reward $r\_{t+k}$ can depend on?**
>
> A: Thanks for asking. In Section 3.1, we specify that $G\_t = \sum^{\infty}\_{k=0} \gamma^k r\_{t+k}$ is the discounted return and $r\_{t+k}$ is the reward computed by $R$ for all agents at $t+k$. Here, the reward function $R : \mathbb{S} \times \mathbb{A} \rightarrow \mathbb{R}$ maps the states and joint actions to real numbers. $R$ can be considered as a function that outputs the immediate reward for each joint action $\mathbf{a} \in \mathcal{A}$. In particular, the reward $r\_{t+k}$ can also be written as $R(s\_{t+k}, \mathbf{a}\_{t+k})$. Therefore, $r\_{t+k}$ depends on the state $s\_{t+k}$ and the joint action of all agents $\mathbf{a}\_{t+k}$.
>
> **2. Page 4, Definition 3.2: Under which assumptions on the model can we guarantee that the Individual-Global-Max (IGM) condition is satisfied?**
>
> A: Thanks for the question. In order to guarantee that the Individual-Global-Max (IGM) condition holds, different assumptions have been made [1, 2, 3].
>
> VDN [1] utilizes a sufficient condition, named additivity, for IGM:
> $$Q\_{tot}(\boldsymbol{\tau}, \mathbf{a}) = \sum^{N}\_{i=1} Q\_i (\tau\_i, a\_i)$$
>
> VDN is able to factorize the joint value function assuming the additivity of individual value functions. Alternatively, QMIX [2] uses a sufficient condition called monotonicity for IGM:
>
> $\frac{\partial Q\_{tot}(\boldsymbol{\tau}, \mathbf{a})}{\partial Q\_i (\tau\_i, a\_i)} \geq 0, \forall i \in $ \{1,...,$N$\}.
>
> IGM is guaranteed under this monotonicity assumption [2].
>
> Lastly, QTRAN [3] proposes to find individual action-value functions $[Q\_i]$ that factorize the original joint action-value function $Q\_{jt}$ and to transform $Q\_{jt}$ into a new value function $Q\_{jt}'$ that shares the same optimal joint action with $Q\_{jt}$. The sufficient condition for $[Q\_i]$ to satisfy the IGM is described in detail in Theorem 1 of [3].

---

> ### Author Response · Authors · 2024-07-18
> **Authors' Response to Reviewer FHth (2)**
>
> **3. Page 8: Why taking a Normal distribution for the reference distribution $p$?**
>
> A: Thanks for the question. As discussed in Section 4.1, the actual conditional distribution of the priors, $p(\hat{s}\_{t}| \hat{s}\_{t-1}, \mathbf{a}\_{t-1})$, is unknown. Inspired by the probabilistic approach in [4], we introduce a new parametrized model, $p\_{\theta}^{\mathrm{prior}}(\hat{s}\_{t}| \hat{s}\_{t-1}, \mathbf{a}\_{t-1})$, to estimate the actual conditional distribution. We construct a variational auto-encoder (VAE) for learning $p\_{\theta}^{\mathrm{prior}}(\hat{s}\_{t}| \hat{s}\_{t-1}, \mathbf{a}\_{t-1})$. In standard VAEs, the prior $p(\cdot)$ can often be set as a standard Gaussian [4]. Therefore, when we calculate the $\mathcal{L}\_{\mathrm{ac}}^{\mathrm{KL}}(\theta)$ loss for the action-conditioned module in Section 4.3, we also need to consider the KL divergence between $p\_{\theta}^{\mathrm{prior}}(\hat{s}\_{t}| \hat{s}\_{t-1}, \mathbf{a}\_{t-1})$ and the standard Gaussian $p(s\_t)$.
>
> **4. Page 9, equation (4): The overall loss contains many terms and it might be difficult to know how to weigh the importance of each term. Do you have guidelines to choose the values of $\beta\_i$? Did you ever obtain very poor results for some “bad” choices of $\beta$'s?**
>
> A: Thanks for the question. The hyper-parameters $\beta\_i$ are in the overall loss function and depend on the three branches in the world model. As suggested by Section 4.1.4, if a branch is masked out, the corresponding $\beta\_i$ will be set to 0.
>
> For choosing the values of $\beta\_i$, we had some hyper-parameter tuning before presenting the final results in the paper. The performance may vary based on the choices of hyper-parameters, but we have never obtained very poor results, partly because we did not set $\beta\_i$ to be unreasonably high. We report the optimal hyper-parameter settings we have obtained so far in Appendix C.
>
> **5. Is the code going to be made available?**
>
> A: Thanks for the inquiry. Yes, we have provided the code in the Supplementary Material of our submission. Furthermore, we will provide a revised version of the manuscript that addresses all the questions proposed by the reviewer. Following the TMLR guidelines, it will be uploaded after all reviewers have submitted their reviews. In the meantime, please let us know if you have any questions regarding our response.
>
> [1] Sunehag, P., Lever, G., Gruslys, A., Czarnecki, W. M., Zambaldi, V., Jaderberg, M., ... \& Graepel, T. (2017). Value-decomposition networks for cooperative multi-agent learning. *arXiv preprint arXiv:1706.05296.*
>
> [2] Rashid, T., Samvelyan, M., Schroeder, C., Farquhar, G., Foerster, J., \& Whiteson, S. (2018). QMIX: Monotonic Value Function Factorisation for Deep Multi-Agent Reinforcement Learning. In *International Conference on Machine Learning* (pp. 4295-4304). PMLR.
>
> [3] Son, K., Kim, D., Kang, W. J., Hostallero, D. E., \& Yi, Y. (2019). Qtran: Learning to factorize with transformation for cooperative multi-agent reinforcement learning. In *International Conference on Machine Learning* (pp. 5887-5896). PMLR.
>
> [4] Huang, S., Su, H., Zhu, J., \& Chen, T. (2020). SVQN: Sequential Variational Soft Q-Learning Networks. In *Eighth International Conference on Learning Representations*.

---

### Review · Reviewer_9ker · 2024-08-15

**Summary Of Contributions:**

This work proposes a new method called Value Decomposition Framework with a Disentangled World Model (VDFD) to improve the sample efficiency and learning performance of MARL algorithms (e.g., QMIX). The key idea is to learn a world model by decoupling the transition dynamics into three components: an action-conditioned branch for controllable dynamics, an action-free branch for non-controllable dynamics, and a static branch for static information. The learned world model is then used to provide imagination rollouts as an additional input to MARL algorithms. The proposed method is evaluated in StarCraft II micro-management, Multi-Agent MuJoCo, and Level-Based Foraging challenges.

**Audience:**

Yes

**Claims And Evidence:**

No

**Requested Changes:**

Please refer to my comments on the “Weaknesses” for my requested changes in writing, organization, and experiments.

**Strengths And Weaknesses:**

### Strengths:

- Leveraging decoupled model-based RL in multi-agent problems is novel to my knowledge.
- The proposed method is evaluated in three types of environments. The ablation study is also provided.

&nbsp;

### Weaknesses:

- From my point of view, it is a bit too late to introduce the key idea of how to leverage the learned world model to facilitate QMIX in Section 4.2. The content in Section 4.1 is basically fairly independent from multi-agent problems (except for the joint observations and actions). I recommend presenting the key concept/idea of the proposed method first and then diving into the details. Besides, Figure 1 and Figure 2 are detailed illustrations of the network architecture, which do help much for the readers to understand how the learned world model is used to facilitate MARL, as I finally found it is done in a way more like data augmentation rather than dyna-style model-based RL.
- The writing is unsatisfactory and somewhat poor. The notations and formulations are not clear, and this directly leads to redundant expressions and hard-to-read text in Section 4. Here are a few concrete ones:
    - I feel a bit ambiguity between the static component $z^{static}$ and the joint observation $z_t$. This caused some difficulty when I read Section 4.1.
    - The notation $s$ for the state of the MA system and $\hat{s}$ for the action-conditioned branch (which corresponds to the state transition) are somewhat confusing. The state $s$ should contain all the information of the controllable part, non-controllable part and the static part. If the aim is to decouple the three components, I recommend using different notations to avoid such kinds of ambiguity.
    - Why the action-conditioned branch does not depend on the non-controllable factor $\hat{v}$? Or the information of the non-controllable factor $\hat{v}$ is contained by $\hat{s}$? If so, the overlap cripples the decoupling here. Moreover, since the non-controllable factor $\hat{v}$ is conditioned on $z$ and $z$ depends on the action made in the previous step, the sense of the action-free/non-controllable factor breaks. the formal modeling for the three decoupled components does not make sense to me.
    - The mixed dynamics $d$ is introduced at the beginning of Section 4.1 once and then mentioned in Section 4.1.4. It is a bit too far.
- The use of the world model learned in this work is different from prior works or the conventional model-based RL principle, e.g., MBRL, Dreamer. The QMIX network only takes the model-based imagination rollouts as additional inputs, thus being more like Data Augmentation rather than typical Model-based RL that uses model-based imagination rollouts as virtual interaction experiences.
- Many additional hyperparameters are introduced, which makes me worry about the practical significance of the proposed method especially if the audience wants to adopt the proposed method for new MA problems:
    - $\beta_1$ to $\beta_5$ are hyper-parameters to balance the individual loss terms in Equation 4.
    - Right above Equation 3, the authors mentioned “Let $M^{ac}$, $M^{af}$, and $M^{st}$ be real-valued masks, which can be viewed as hyper-parameters”. I did not find how these masks are selected or decided in the paper, which is confusing. (Or the masks are only used for the ablation?)
- Some related works are missing [1-5]. I recommend the authors to include them in the related work.
- The experimental results are not very convincing.
    - In Figure 3 of this paper, QMIX totally failed in MMM2, 2c_vs_64zg, 27m_vs_30m. The performance of baseline method QMIX is much lower than the performance reported in prior works (e.g., Figure 5 in [5]) which also uses SMAC for evaluation.
    - I recommend to include HPN-QMIX [5] as a baseline.
    - For MAMuJoCo, the authors mentioned “MAMuJoCo also introduces partial observability, as the agents can be configured with different levels of observational capabilities”. However, the concrete level of partial observability is not given in the main body of the paper.
    - Are the hyperparameters reported in Table 7 used for all the environments considered in the experiment section? I found the text under Table 7 is a bit ambiguous and the authors need to make it clearer.

&nbsp;

### Minor points

- The organization should be improved, e.g., there are four subsubsection (some are too short) in Section 4.1.
- The formats of the papers in Reference are inconsistent, e.g., for the arxiv papers.

---

Reference:

[1] Model-based Multi-agent Reinforcement Learning: Recent Progress and Prospects. arXiv:2203.10603

[2] Efficient model-based multi-agent mean-field reinforcement learning. 2021.

[3] Scalable multi-agent model-based reinforcement learning. 2022

[4] Efficient model-based multi-agent reinforcement learning via optimistic equilibrium computation. ICML 2022.

[5] Boosting Multiagent Reinforcement Learning via Permutation Invariant and Permutation Equivariant Networks. ICLR 2023

---

> ### Author Response · Authors · 2024-08-26
> **Authors' Response to Reviewer 9ker (1)**
>
> Dear reviewer,
>
> We genuinely appreciate your valuable feedback. Below, you will find our response to your comments and queries. In accordance with the TMLR guidelines, we will submit a revised manuscript addressing the concerns of all reviewers soon.
>
> **Weaknesses and Requested Changes:**
>
> **1. It's a bit late to introduce the key idea of how to leverage the learned world model to facilitate QMIX in Section 4.2. I recommend presenting the key concept/idea of the proposed method first and then diving into the details. Besides, Figure 1 and Figure 2 are detailed illustrations of the network architecture.**
>
> A: Thanks for the advice. In our updated manuscript, we will revise Section 4. We will present an overview of the key ideas of our VDFD method at the beginning of the section, followed by detailed descriptions in the sub-sections.
>
> Regarding the figures in the main paper, we will also modify Figure 1 and Figure 2 to further enhance their clarity and make them easier to understand. The new figures will better illustrate the architectures of the generative models and the framework of VDFD.
>
> **2. The writing is unsatisfactory. There is a bit of ambiguity between the static component $z^{static}$ and the joint observation $z\_t$.**
>
> A: Thanks for pointing out. Yes, in the paper we denote the agents' observations as $\mathbf{z}$ and the joint observation at time $t$ as $\mathbf{z}\_{t}$. Meanwhile, in the static branch of the world model, we denote the output of the variational graph auto-encoder GALA as $\mathbf{z}^\mathrm{static}$, which is also displayed in Figure 1. We will make a note in Section 4.1 to clarify the ambiguity.
>
> **3. The notation $s$ for the state of the MA system and $\hat{s}$ for the action-conditioned branch are somewhat confusing. The state $s$ should contain all the information of the controllable part, the non-controllable part and the static part.**
>
> A: Thanks for pointing out the ambiguity in these notations. In our paper, the global state is denoted as $s$, which embodies all the information being used in the action-conditioned, action-free, and static branches of the world model. On the other hand, we define the action-conditioned (controllable) latent state as $\hat{s}$, which is used for the state transitions in the action-conditioned branch. We will specify this in Section 4 of the updated manuscript to resolve the ambiguity.

---

> ### Author Response · Authors · 2024-08-26
> **Authors' Response to Reviewer 9ker (2)**
>
> **4. Why the action-conditioned branch does not depend on $\hat{v}$? Is the information of $\hat{v}$ contained by $\hat{s}$? Moreover, since $\hat{v}$ is conditioned on $z$ and $z$ depends on the action made in the previous step, the sense of the action-free/non-controllable factor breaks.**
>
> A: Thanks for the queries. In the proposed model, the non-controllable (action-free) branch serves the purpose of identifying passive forces (i.e., $\hat{v}$) surrounding the agents. This branch can also be described as modeling the dynamics of the environment that are not regulated by the agent's actions. It is true that the non-controllable dynamics $\hat{v}$ may still affect the controllable dynamics $\hat{s}$. For example, in the training phase of a competitive multi-agent environment, such as StarCraft II, the non-controllable branch collects the observations of opponents as data. The attack of the opponents may force the agents to change their strategy to reduce damage. We have considered this possibility, and therefore we proposed that the past information of the environment, $\mathbf{h}\_{t}$, be the input to both the GALA model and the agent network in the action-conditioned branch. By definition, $\mathbf{h}\_{t}$ encapsulates the non-controllable information as well. During action-conditioned state transition, GALA outputs $\hat{s}\_{t} \sim q\_{\theta}(\cdot | \mathbf{h}\_t )$ and the agent network outputs $\mathbf{a}\_{t}' \sim \pi(\cdot | \boldsymbol{\tau}\_{t} )$. Both $\hat{s}\_{t}$ and $\mathbf{a}\_{t}'$ depend on $\mathbf{h}\_{t}$, which indeed encapsulates the non-controllable factor $\hat{v}$.
>
> For the question regarding $\hat{v}$ and $z$, it is correct that $z$ can be influenced by previous actions. For example, in a StarCraft II environment, agents can fire at an enemy and then observe that the enemy is dead. The other enemies are likely to react to its death and revise their strategy, which is the reason why $\hat{v}$ is conditioned on $z$. Taking this into consideration, we have specified in Section 4.1.4 that the boundary between action-free and action-conditioned components might be unclear in complicated multi-agent systems. Learning to perfectly disentangle the dynamics in such systems has always been a challenging research topic. Our approach to this challenge is to follow a clear design principle, as stated in 4.1.4. With this approach, we can decouple the environment dynamics into modular representations using a unified disentangled framework even in complex systems.
>
> We will address these questions and improve the clarity of writing in the updated manuscript.
>
> **5. The mixed dynamics $d$ is introduced at the beginning of Section 4.1 once and then mentioned in Section 4.1.4. It is a bit too far.**
>
> A: Thanks for pointing out. We can add a note at the beginning of Section 4.1 that we will use this concept again in Section 4.1.4 when we discuss the modules of the world model.
>
> **6. The QMIX network only takes the imagination rollouts as additional inputs, thus being more like Data Augmentation rather than typical Model-based RL.**
>
> A: Thanks for the comment. Yes, we agree that in principle, our approach shares more similarities with methods that augment model-free RL using imagined data from world models than with methods that train policies directly with model-based imaginary roll-outs. To complement this, we will provide a more extensive discussion of similar works such as [6] in the Related Work of our updated manuscript.
>
> **7. Many additional hyperparameters are introduced such as the $\beta$'s, which could affect the practical significance of the method if adopted for new MA problems.**
>
> A: Thanks for the comment. We will add a section on limitations and broader impact in the appendix of the updated manuscript and include this point. We agree that hyper-parameter tuning may be needed when our method is applied to new multi-agent environments. For example, in order to choose the values of $\beta\_i$, we had some hyper-parameter tuning before presenting the final results in the paper. The performance may vary based on the choices of hyper-parameters, but we did not obtain very poor results, and the reason could be that we did not set the hyper-parameters to extreme values.

---

> ### Author Response · Authors · 2024-08-26
> **Authors' Response to Reviewer 9ker (3)**
>
> **8. It's confusing how these real-valued masks are selected or decided in the paper, or if they are only used for ablation.**
>
> A: Thanks for the comment. $\mathbf{M}^{\mathrm{ac}}, \mathbf{M}^{\mathrm{af}}, \mathbf{M}^{\mathrm{st}}$ are real-valued masks that are pre-defined without training. They are used for the transformation of the outputs of the modularized world model. In Section 4.1.4, the outputs of the action-conditioned, action-free, and static branches are denoted as $\mathbf{o}^{\mathrm{ac}}$, $\mathbf{o}^{\mathrm{af}}$, and $\hat{\mathbf{z}}^{\mathrm{static}}$, respectively. Moreover, Section 4.2 indicates that the transformation of the outputs of the three branches can be written as $\hat{s\_{t}}^{\mathrm{Rollout}} = \mathbf{M}^{\mathrm{ac}} \odot \mathbf{o}^{\mathrm{ac}}$, $\hat{v\_{t}}^{\mathrm{Rollout}} = \mathbf{M}^{\mathrm{af}} \odot \mathbf{o}^{\mathrm{af}}$, and $\hat{\mathbf{z}}^{\mathrm{Rollout}} = \mathbf{M}^{\mathrm{st}} \odot \hat{\mathbf{z}}^{\mathrm{static}}$. Therefore, if we would like to mask out one of the branches in the world model (e.g., in the ablation studies), the corresponding mask will be set to zero.
>
> Empirically, we had some hyper-parameter tuning to choose the values of the masks within the range of [0, 1]. In Section 5, we have specified that $\beta\_1$ to $\beta\_5$ are hyper-parameters corresponding to the real-valued masks. In practice, we chose the value of $\mathbf{M}^{\mathrm{ac}}$ to be $1/2$, $\mathbf{M}^{\mathrm{af}}$ to be $1/2$, and $\mathbf{M}^{\mathrm{st}}$ to be $1/3$, consistent with the values of the $\beta$'s reported in Appendix C.
>
> We will clarify this question in the updated manuscript.
>
> **9. Some related works are missing [1-5]. I recommend the authors to include them.**
>
> A: Thanks for providing additional related works. We will revise Section 2 and add the discussion on these papers.
>
> **10. The performance of QMIX is much lower than the performance reported in prior works (e.g., Figure 5 in [5]).**
>
> A: Thanks for the question. We believe the reason is that the authors in [5] adopted the optimized QMIX proposed in [7], which is specified in Section 5.1 of [5]. In contrast, we used the original QMIX proposed in [8], as specified in Section 5 of our manuscript. There is a large performance gap between the original QMIX and the optimized variant, as demonstrated in [7]. This explains why the baseline QMIX in our experiments showed much lower performance compared to the baseline in [5].
>
> **11. It's recommended to include HPN-QMIX [5] as a baseline.**
>
> A: Thanks for the advice. Given the timeline of the TMLR reviewing process, we will try to run experiments on HPN-QMIX and obtain the results by the end of August. We will include the results of HPN-QMIX in the updated appendix.
>
> **12. The concrete level of partial observability for MAMuJoCo is not given in the main body of the paper.**
>
> A: Thanks for pointing out. Yes, the level of partial observability for MAMuJoCo is given in the appendix but we will specify it in the updated main paper.
>
> **13. Are the hyperparameters reported in Table 7 used for all the environments considered in the experiment section?**
>
> A: Thanks for asking. We will improve the clarity of the description about Table 7. The hyper-parameters in Table 7 are indeed used for all the benchmark environments, with the exceptions that the number of time steps for MAMuJoCo was set to 7M and the number of time steps for the ablation experiments in the *MMM2* environment (Section 5.4) was set to 3M, so that the performance of the ablations can plateau.
>
> **14. Minor points regarding organization and references.**
>
> A: Thanks for the suggestions. We will try to address them in the updated manuscript.
>
> Please let us know if you have any questions regarding our response.

---

> ### Author Response · Authors · 2024-08-26
> **Authors' Response to Reviewer 9ker (4)**
>
> **References**
>
> [1] Model-based Multi-agent Reinforcement Learning: Recent Progress and Prospects. *arXiv:2203.10603*
>
> [2] Efficient model-based multi-agent mean-field reinforcement learning. 2021.
>
> [3] Scalable multi-agent model-based reinforcement learning. 2022
>
> [4] Efficient model-based multi-agent reinforcement learning via optimistic equilibrium computation. ICML 2022.
>
> [5] Boosting Multiagent Reinforcement Learning via Permutation Invariant and Permutation Equivariant Networks. ICLR 2023.
>
> [6] Racaniere, S., Weber, T., Reichert, D., Buesing, L., Guez, A., Jimenez Rezende, D., ... \& Wierstra, D. (2017). Imagination-augmented agents for deep reinforcement learning. *Advances in neural information processing systems,* 30.
>
> [7] Hu, J., Jiang, S., Harding, S. A., Wu, H., \& Liao, S. W. (2021). Rethinking the implementation tricks and monotonicity constraint in cooperative multi-agent reinforcement learning. *arXiv preprint arXiv:2102.03479*.
>
> [8] Rashid, T., Samvelyan, M., Schroeder, C., Farquhar, G., Foerster, J., \& Whiteson, S. (2018). QMIX: Monotonic Value Function Factorisation for Deep Multi-Agent Reinforcement Learning. In *International Conference on Machine Learning* (pp. 4295-4304). PMLR.

---

> > ### Comment · Reviewer_9ker · 2024-09-12
> >
> > I appreciate the authors for providing detailed responses and extra experiments, as well as the great efforts in revising the paper.

---

### Author Response · Authors · 2024-08-28
**Submission of the Revised Paper by Authors**

Dear reviewers,

We sincerely appreciate your time and valuable feedback on our paper. We are writing to kindly inform you that we have submitted the revised version of our paper, incorporating the suggestions provided by all three reviewers.
Below is a brief summary of the changes made in the revised paper:

- **Figure 1**: Revised following Reviewer mLpZ’s suggestions.

- **Figure 2**: Revised following Reviewer mLpZ’s suggestions. It has been divided into two figures: an illustration of the disentangled roll-outs and an illustration of the overall framework that we proposed.

- **Section 2, Related Work**: Revised following the suggestions of Reviewers 9ker and mLpZ.

- **Section 3, Background**: Revised following the suggestions of Reviewers FHth and mLpZ. Added a new subsection (Section 3.3) to introduce the concepts of structured variational inference and variational lower bound.

- **Section 4, Method**: Revised following the suggestions of all three Reviewers.

- **Section 5, Experiments**: Revised following the suggestions of Reviewers 9ker and mLpZ.

- **Section 6, Conclusion**: Included the intuitive visualization of disentanglement learning as a future direction.

- **Appendix B, Supplemental Results**: Revised following the suggestions of Reviewers 9ker and mLpZ. Appendix B.6 (Comparison Between VDFD and HPN-QMIX) is a newly added subsection, following Reviewer 9ker’s recommendation.

- **Appendix C, Experimental Details**: Revised following Reviewer 9ker’s  suggestions.

- **Appendix G, Limitations and Broader Impact**: Newly added following the suggestions of Reviewers FHth and 9ker.

We look forward to your further feedback. Thank you for your consideration.

---

### Decision · Action_Editor_QPwj · 2024-11-07

**Recommendation:** Reject

**Comment:**

This manuscript has undergone comprehensive reviewing and two rounds of revising. The conclusions reached by the three reviewers are one Weak Accept, one Weak Reject, and one Reject. Reviewers are positive about the novelty and relevance of this work, but are negative about the empirical evaluation and presentation quality. Authors failed to convince the reviewers in this constructive editorial process. To highlight, the proposed solution lacks elaboration on the decoupled modeling, simplicity for practical adoption, and superiority in empirical performance.

**Audience:**

Yes. World models are of wide interests in the reinforcement learning community, while extending this series of models to multiagent learning systems is of relevance.

**Claims And Evidence:**

No. For example, according to the extra experiments provided by the authors in Appendix B.6, HPN-QMIX seems to show an overall better performance than the proposed method VDFD.